# Redox nanomedicine ameliorates chronic kidney disease (CKD) by mitochondrial reconditioning in mice

Aniruddha Adhikari [1], Susmita Mondal[1], Tanima Chatterjee[2], Monojit Das[3,4], Pritam Biswas[5], Ria Ghosh[2], Soumendra Darbar[6], Hussain Alessa[7], Jalal T. Althakafy[7], Ali Sayqal[7], Saleh A. Ahmed[7,8], Anjan Kumar Das[9], Maitree Bhattacharyya[2] & Samir Kumar Pal [1,3 ✉]

Targeting reactive oxygen species (ROS) while maintaining cellular redox signaling is crucial in the development of redox medicine as the origin of several prevailing diseases including chronic kidney disease (CKD) is linked to ROS imbalance and associated mitochondrial dysfunction. Here, we have shown that a potential nanomedicine comprising of $Mn_3O_4$ nanoparticles duly functionalized with biocompatible ligand citrate (C-$Mn_3O_4$ NPs) can maintain cellular redox balance in an animal model of oxidative injury. We developed a cisplatin-induced CKD model in C57BL/6j mice with severe mitochondrial dysfunction and oxidative distress leading to the pathogenesis. Four weeks of treatment with C-$Mn_3O_4$ NPs restored renal function, preserved normal kidney architecture, ameliorated over-expression of pro-inflammatory cytokines, and arrested glomerulosclerosis and interstitial fibrosis. A detailed study involving human embryonic kidney (HEK 293) cells and isolated mitochondria from experimental animals revealed that the molecular mechanism behind the pharmacological action of the nanomedicine involves protection of structural and functional integrity of mitochondria from oxidative damage, subsequent reduction in intracellular ROS, and maintenance of cellular redox homeostasis. To the best of our knowledge, such studies that efficiently treated a multifaceted disease like CKD using a biocompatible redox nano-medicine are sparse in the literature. Successful clinical translation of this nanomedicine may open a new avenue in redox-mediated therapeutics of several other diseases (e.g., diabetic nephropathy, neurodegeneration, and cardiovascular disease) where oxidative distress plays a central role in pathogenesis.

[1] Department of Chemical, Biological and Macromolecular Sciences, S. N. Bose National Centre for Basic Sciences, Kolkata, India. [2] Department of Biochemistry, University of Calcutta, Kolkata, India. [3] Department of Zoology, Uluberia College, University of Calcutta, Uluberia, Howrah, India. [4] Department of Zoology, Vidyasagar University, Rangamati, Midnapore, India. [5] Department of Microbiology, St. Xavier's College, Kolkata, India. [6] Research & Development Division, Dey's Medical Stores (Mfg.) Ltd, Kolkata, India. [7] Department of Chemistry, Faculty of Applied Sciences, Umm Al-Qura University, Makkah, Saudi Arabia. [8] Chemistry Department, Faculty of Science, Assiut University, Assiut, Egypt. [9] Department of Pathology, Calcutta National Medical College and Hospital, Kolkata, India. ✉email: skpal@bose.res.in

Reactive oxygen species (ROS) have long been considered as an unwanted but inevitable byproduct of aerobic oxygen metabolism[1]. Excessive generation of ROS may lead to tissue damage and numerous undesired physiological consequences. Increased ROS level is linked to inflammation, aging, and pathogenesis of diseases like diabetes, cancer, atherosclerosis, chronic kidney disease (CKD), and neurodegeneration[2–5]. Recent understanding about the pivotal role of ROS as secondary messengers in cellular signaling to control processes like metabolism, energetics, cell survival, and death lead to a paradigm shift to the traditional "oxidants are bad —antioxidants are good" based simplistic view of redox biology[6–10]. Lack of attention towards the paradox between lethality of excessive intracellular ROS (oxidative distress) and the beneficial role of low concentration ROS (oxidative eustress) is the major underlying reason behind the failure of conventional antioxidant therapies using natural or synthetic antioxidants (e.g., α-tocopherol, ascorbic acid, β-carotene, curcumin, and numerous dietary polyphenols) that along with stoichiometric scavenging of intracellular free radicals, insulate redox signaling[10–12]. Moreover, meta-analyses of clinical trials show that the conventional antioxidants are not only ineffective, but also harmful, and even increase mortality[12,13]. The understanding that proper cell functioning critically requires a dynamic balance between oxidative eustress and distress (i.e., cellular redox homeostasis) forms the conceptual framework of redox medicine, a novel therapeutics that passivates the oxidative distress while maintaining the normal redox circuitry[10,12,14–16]. The cellular redox dynamics and its regulations, however, are still largely elusive because of the lack of effective pharmacological interventions[17]. In this regard, biocompatible transition metal oxide nanoparticles with potential electron-donating as well as accepting capability could be a viable option provided they are stable in the biological system, able to assimilate in the targeted tissue, and function in the physiological milieu.

Recently, we have shown that spinel structured citrate functionalized $Mn_3O_4$ nanoparticles (C-$Mn_3O_4$ NPs) have the unique ability to generate ROS in dark, and when injected into jaundiced animals can selectively degrade bilirubin (i.e., a toxic byproduct of heme metabolism) without showing adverse effects to other blood parameters[18]. At the same time, we found that the nanoparticles can catalytically scavenge free radicals particularly $H_2O_2$ in the in vitro reaction system. The microenvironment-controlled (i.e., presence of ROS, and subsequent changes in pH and dissolved $O_2$) dynamic equilibrium between disproportionation and comproportionation involving surface $Mn^{3+}$, $Mn^{4+}$, and $Mn^{2+}$ charge states present in the hausmannite structure of C-$Mn_3O_4$ NPs is responsible for such contrasting activity[19–21]. Therefore, depending upon the intracellular redox condition and pH (which can vary between intra- and extra-cellular environments, within the organelles and subsequently affect the redox activity of the nanoparticles), the nanoparticle has the potential to balance the oxidative distress and eustress, the most important feature of a redox medicine.

In this study, our major aim was to evaluate the potential of C-$Mn_3O_4$ NPs as a redox medicine against CKD. CKD, the progressive decline in kidney function, is one of the most serious global public health problem (with 8–16% worldwide prevalence) that originates from redox imbalance due to mitochondrial dysfunction and have no effective medication till date[22–26]. In order to understand the therapeutic potential of C-$Mn_3O_4$ NPs we used a cisplatin-induced C57BL/6j mice model of CKD. The mechanistic details of their pharmacological action in the maintenance of redox homeostasis and mitoprotection were further explored using cellular (human embryonic kidney cell, HEK 293) as well as animal model.

## Results

### Designing aqueous soluble C-$Mn_3O_4$ NPs to target kidney cells.
The size, surface charge, and surface functionalization ligands determine the biodistribution of a nanomaterial inside living organisms. Protein corona (i.e., proteins adsorbed from plasma or intracellular fluids to the nanoparticle surface) is another important factor that critically influences the in vivo biodistribution and cellular internalization of nanoparticles[27]. Earlier studies have reported that particles with less than 8 nm diameters having moderate to high surface negative charge tend to accumulate in the renal system[28]. Therefore, care was taken at the time of synthesis to control the size of the $Mn_3O_4$ nanoparticles within the range of 6 nm. The transmission electron micrograph (TEM) of C-$Mn_3O_4$ NPs shows the monomodal distribution of nearly spherical particles with an average diameter of $5.58 \pm 2.42$ nm (Fig. 1a). High resolution (HR) TEM image of a single nanoparticle confirms the crystalline nature with clear atomic lattice fringe spacing of $0.311 \pm 0.02$ nm (Fig. 1a-inset) corresponding to the separation between (112) lattice planes of hausmannite $Mn_3O_4$ crystal. All x-ray diffraction (XRD) peaks corresponding to (101), (112), (200), (103), (211), (004), (220), (204), (105), (312), (303), (321), (224), and (400) planes of C-$Mn_3O_4$ NPs (Fig. 1b) exactly reflect the tetragonal hausmannite structure of $Mn_3O_4$ with a lattice constant of a = 5.76 Å and c = 9.47 Å and space group of $I41/amd$ described in the literature (JCPDS No. 24-0734). The absence of any additional peak from other phases indicates the high purity of the synthesized material.

Surface functionalization with carboxyl rich ligand trisodium citrate not only made the nanoparticles biocompatible and aqueous soluble but also helped the surface charge to be negative (i.e., zeta potential, $\xi = -12.23 \pm 0.6$ mV with electrophoretic mobility $-0.96 \pm 0.05 \, \mu$ cm V$^{-1}$ s). Fourier transformed infrared (FTIR) spectroscopy was used to confirm the binding of citrate to the surface of the nanomaterial (Fig. 1c). Broadening of the 630, 514, and 413 cm$^{-1}$ bands associated with stretching vibrations of Mn–O and Mn–O–Mn bonds of $Mn_3O_4$ NPs along with substantial disruption of both symmetric (1410 cm$^{-1}$) and asymmetric (1619 cm$^{-1}$) stretching modes of carboxylates (COO$^-$) of citrate indicates a strong covalent interaction between them.

Previously we showed that C-$Mn_3O_4$ NPs can selectively degrade bilirubin without affecting other blood parameters[28]. Here, initially, we evaluated their potential to scavenge $H_2O_2$ in an in vitro system using Rose Bengal (RB) degradation assay. RB has a distinct absorption peak at 540 nm. In the presence of $H_2O_2$, degradation of RB takes place causing a decrease in the 540 nm absorbance. When added to the reaction mixture, C-$Mn_3O_4$ NPs efficiently prevented the RB from $H_2O_2$ mediated degradation (Supplementary Fig. S1) indicating its strong radical scavenging potential towards $H_2O_2$.

### C-$Mn_3O_4$ NPs maintain redox balance in HEK 293 cells against $H_2O_2$-induced oxidative distress.
In order to test the ability of C-$Mn_3O_4$ NPs to combat oxidative stress in the cellular milieu, we used a cell-based approach. The HEK 293 cells pretreated with different concentrations of nanoparticles (3.75 to 60 µg mL$^{-1}$) were exogenously exposed to $H_2O_2$ (100 µM) and cell viability was estimated using a well-known 2-(4,5-dimethylthiazol-2-yl)-2,5-diphenyltetrazolium bromide (MTT) assay (Fig. 2a). The survival rate for $H_2O_2$ treated cells was ~35% ($p < 0.001$ compared to control, one-way ANOVA, $F_{(11, 48)} = 136.7$). The C-$Mn_3O_4$ NPs protected the cells from $H_2O_2$ induced cell death in a dose-dependent manner. Cell viability reached a maximum of ~85 and ~88% ($p < 0.001$ compared to $H_2O_2$ treated cells, One-way ANOVA, $F_{(11, 48)} = 136.7$) in $H_2O_2$ exposed cells when

pretreated with 30 and 60 µg mL$^{-1}$ NPs, respectively. Pretreatment of the cells with similar concentrations of the NPs alone did not cause significant cellular mortality except the 60 µg mL$^{-1}$ (~18%, $p < 0.001$ compared to control, one-way ANOVA, $F(11, 48) = 136.7$). Based on the results, we selected the 30 µg mL$^{-1}$ C-Mn$_3$O$_4$ NPs for further experiments. Identical results were observed in the lactate dehydrogenase (LDH) assay (Fig. 2b). The presence of a high concentration of H$_2$O$_2$ inside the cell caused oxidative damage to the plasma membrane resulting in an increased release of LDH, a cytosolic enzyme, into the surrounding cell culture medium. Pretreatment with C-Mn$_3$O$_4$ NPs protected the cells from H$_2$O$_2$ induced oxidative damage resulting in a ~40% reduction in the LDH release ($p < 0.001$ compared to H$_2$O$_2$ treated cells, one-way ANOVA, $F(3, 16) = 132.1$). In post hoc analysis, the cells treated with C-Mn$_3$O$_4$ NPs alone also showed significant difference when compared to untreated control ($p = 0.0057$, one-way ANOVA, $F(3, 16) = 132.1$). However, it was not reflected in cell mortality. To evaluate the scavenging of H$_2$O$_2$ by C-Mn$_3$O$_4$ NPs under stress conditions, we monitored the intracellular oxidative stress using a ROS-sensitive fluorescence probe, dihydro dichloro-fluorescein diacetate (DCFH2-DA). DCFH2-DA is transported across the cell membrane and hydrolyzed by intracellular esterases to form nonfluorescent 2′,7′-dichlorofluorescein (DCFH), which is rapidly converted to highly fluorescent 2′,7′-dichlorofluorescein (DCF) in presence of ROS. The results illustrate that H$_2$O$_2$ exposure caused a substantial increase in the cellular ROS level indicated by the enhanced relative green fluorescence ($\lambda_{em/DCFH2-DA} = 520$ nm) intensity of DCFH2-DA ($p < 0.001$ compared to control, one-way ANOVA, $F(3, 16) = 307.6$) when measured using fluorescence microscopy (Fig. 2c, d) or flow cytometry (Fig. 2e). However, pretreatment with 30 µg mL$^{-1}$ C-Mn$_3$O$_4$ NPs significantly lowered intracellular ROS level which was reflected in decreased fluorescence (~50% reduction; $p < 0.001$ compared to H$_2$O$_2$ treated cells, one-way ANOVA, $F(3, 16) = 307.6$) of the probe. The C-Mn$_3$O$_4$ NP treated cells also show a significant amount of ROS ($p < 0.001$ compared to H$_2$O$_2$ treated cells, one-way ANOVA, $F(3, 16) = 307.6$), which may be due to the inherent ability of the nanoparticles to generate ROS. The morphological observations in differential interference contrast (DIC) microscopy (Fig. 2d) support the results of cell viability and oxidative damage evaluation studies. The cells pretreated with C-Mn$_3$O$_4$ NPs prevented the shrinkage and congregation of the cell body due to H$_2$O$_2$ overexposure and maintained normal cellular architecture.

The biological consequences of exposure to nanomaterials can only be understood in terms of content. Therefore, we evaluated the amount of nanoparticles internalized by the cells in terms of cellular manganese content using inductively coupled plasma atomic emission spectroscopy (ICP-AES). Figure 2f shows the dose-dependent uptake of C-Mn$_3$O$_4$ NPs in HEK 293 cells. The amount of intracellular manganese content (i.e., the nanoparticle content) increased logistically with the increase in the administered dose range of 3.75 to 60 µg mL$^{-1}$, and reached a plateau at ~50 µg mL$^{-1}$. Figure 2g depicts the correlation between the biological impacts (as measured by MTT and ROS assays) with both administered dose and intracellular nanoparticle content. Both prevention of cell death and quenching of intracellular ROS followed a logistic (Hill equation: $y = A_1 + \frac{A_2 - A_1}{1 + 10^{(\log x_0 - x)p}}$) relationship with administered dose and intracellular nanoparticle content.

**C-Mn$_3$O$_4$ NPs prevent mitochondria, the master redox regulator, from H$_2$O$_2$-induced oxidative damage.** Mitochondria despite being the primary source and regulator of intracellular ROS,

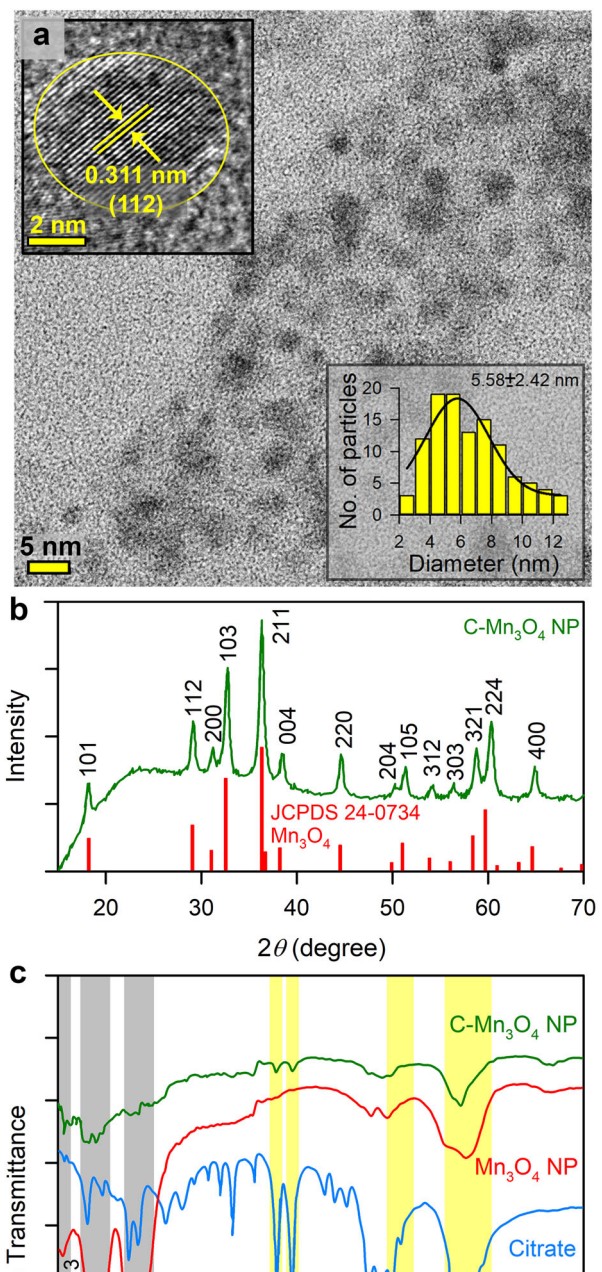

**Fig. 1 Characterization of C-Mn$_3$O$_4$ NPs. a** TEM image of C-Mn$_3$O$_4$ NPs shows the spherical shape of the nanoparticles with the monomodal distribution. Inset shows an HRTEM image of a single nanoparticle having a high crystalline structure with 0.311 nm interfringe distance corresponding to the (112) plane. The other inset shows the histogram for the size distribution of the nanoparticles having an average diameter of 5.58 ± 2.42 nm. **b** Experimental XRD peaks of the nanoparticle exactly match that of Mn$_3$O$_4$ hausmannite defined in the literature (JCPDS No. 24-0734). **c** FTIR spectra of C-Mn$_3$O$_4$ NPs, Mn$_3$O$_4$ NPs, and citrate. Perturbation at Mn–O stretching at 413, 514, 630 cm$^{-1}$ (shaded gray) of Mn$_3$O$_4$ NPs and carboxylic groups at 1066, 1112, 1410, 1619 cm$^{-1}$ (shaded yellow) of citrate confirms strong covalent binding between citrate and the nanoparticle.

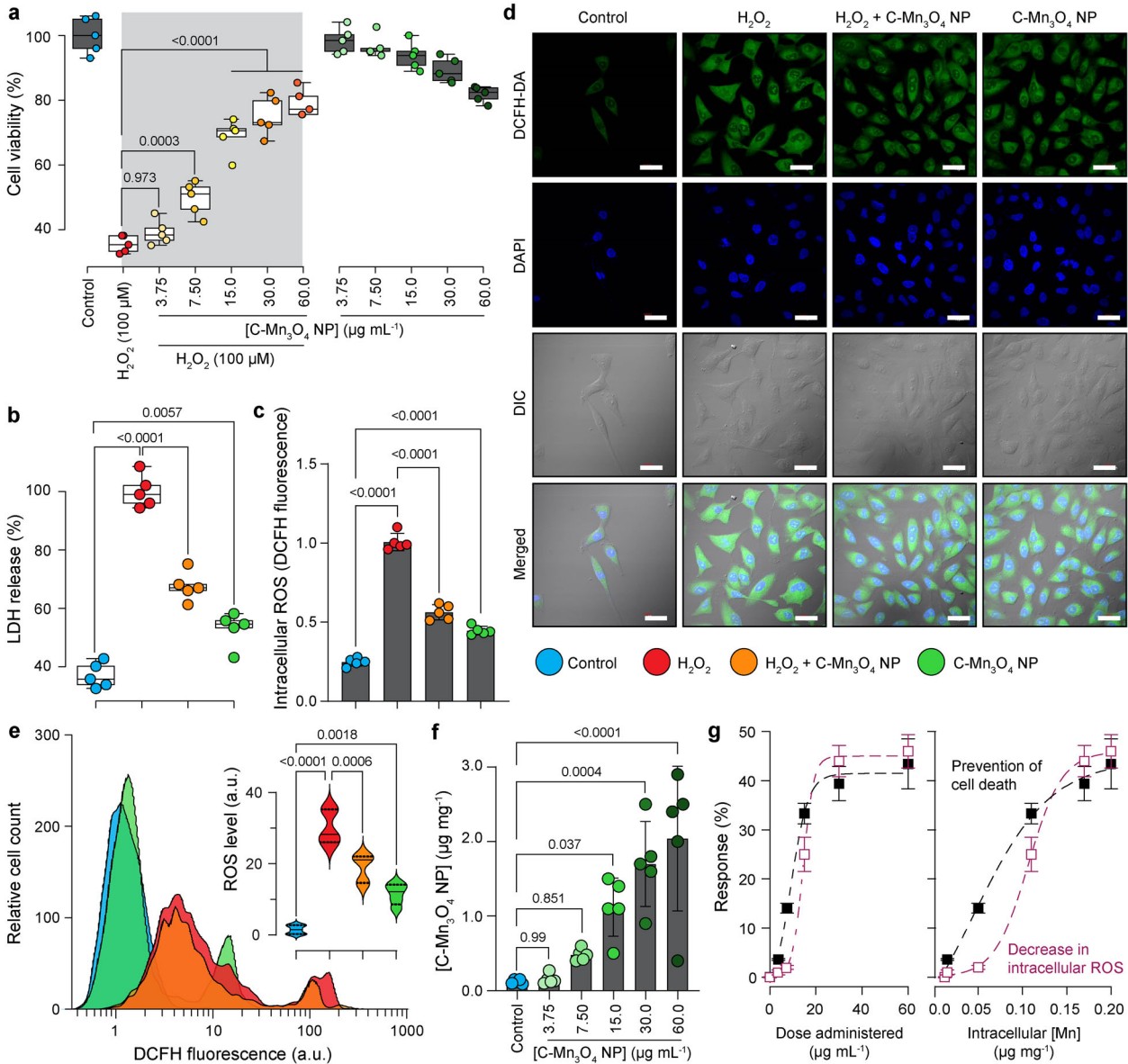

**Fig. 2 Ability of C-Mn$_3$O$_4$ NPs in scavenging of intracellular ROS. a** Cell viability was measured using MTT. The gray shaded area represents H$_2$O$_2$ treatment. **b** LDH release. **c** Quantification of intracellular ROS as estimated from DCF fluorescence observed under confocal microscopy. **d** Confocal fluorescence micrographs of HEK 293 cells stained with DCFH2-DA and counterstained with DAPI. Cells were either left untreated or pretreated with C-Mn$_3$O$_4$ NPs (30 μg mL$^{-1}$) prior exposure to H$_2$O$_2$ (100 μM). Scale bar: 30 μm. **e** Flow cytometry of HEK 293 cells stained with DCFH2-DA. Inset—intracellular ROS level quantified from flow cytometry analysis. **f** Dose-dependent internalization of C-Mn$_3$O$_4$ NPs in HEK 293 cells. The intracellular nanoparticle concentration as measured using inductively coupled plasma atomic emission spectroscopy (ICP-AES). **g** Correlation between biological impact (cell death and scavenging of ROS) and administered dose or intracellular nanoparticle content. All four nanoparticle concentration-biological response data are fitted with the Hill equation: $y = A_1 + \frac{A_2 - A_1}{1 + 10^{(\log x_0 - x)p}}$. In bar plots data were expressed as Mean ± SD. In box plots, center lines show the medians; box limits indicate the 25th and 75th percentiles, whiskers extend 1.5 times the interquartile range from the 25th and 75th percentiles. Violins depict kernel density estimation of the underlying data distribution with the width of each violin scaled by the number of observations at that Y-value. Three lines (from the bottom to the top) in each violin plot show the location of the lower quartile (25th), the median, and the upper quartile (75th), respectively. The shaded area indicates the probability distribution of the variable. Individual data points are represented as colored circles ($N = 5$). One-way analysis of variance (ANOVA) followed by Tukey's post hoc multiple comparison test was performed for comparison among the groups. The numbers inside the plots indicate numerical $p$ values. $p < 0.05$ is considered significant.

are the most susceptible organelle to oxidative damage leading to redox imbalance and cell death[29,30]. So, to get further insight into the free radical scavenging activity of C-Mn$_3$O$_4$ NPs we evaluated their protective effect towards mitochondria. Treatment with H$_2$O$_2$ drastically decreased the mitochondrial membrane potential ($\Delta\Psi_m$), as indicated by enhanced rhodamine 123 (Rh123) fluorescence (Fig. 3a, b) along with a burst in mitochondrial ROS production, as

indicated by the increased fluorescence of Mito-sox red (Fig. 3a, c). Pretreatment with 30 μg mL$^{-1}$ C-Mn$_3$O$_4$ NPs significantly restored the $\Delta\Psi_m$ and reduced the mitochondrial ROS. $\Delta\Psi_m$ has a causal relationship with mitochondrial permeability transition pore (mPTP) opening. The results of the Ca$^{2+}$ induced mitochondrial swelling assay indicated that the NPs were effective in preventing the H$_2$O$_2$ induced mPTP opening (Fig. 3d), therefore, maintaining

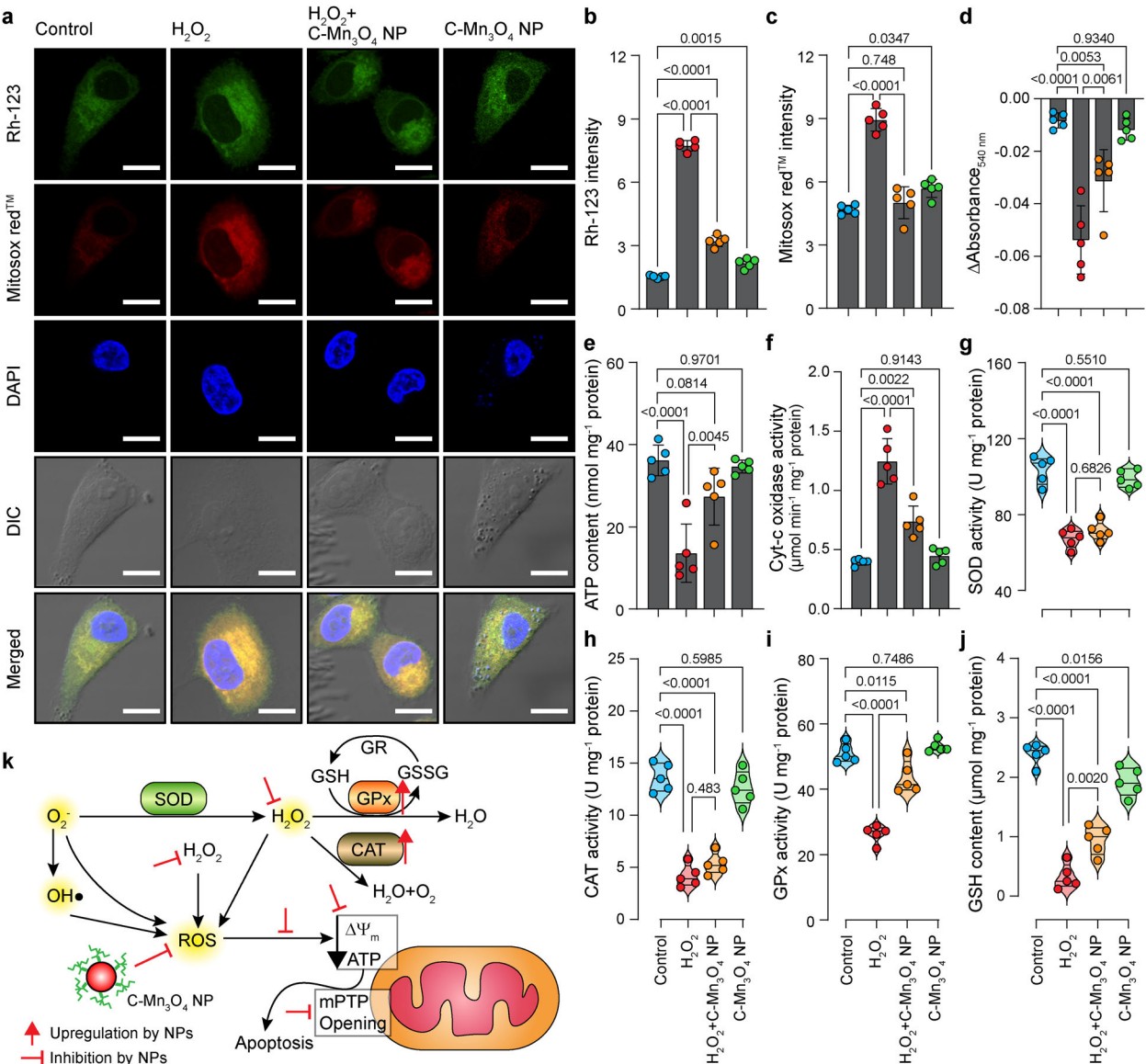

**Fig. 3 Potential of C-Mn₃O₄ NPs in the regulation of cellular redox condition and protection of mitochondria from oxidative damage. a** Confocal fluorescence micrographs of HEK 293 cells stained with rhodamine 123, Mitosox™ red, and counterstained with DAPI. Cells were either left untreated or pretreated with C-Mn₃O₄ NPs (30 µg mL⁻¹) prior exposure to H₂O₂ (100 µM). Scale bar: 10 µm. **b** Intensity of rhodamine 123 as a marker of mitochondrial membrane potential ($\Delta\Psi_m$). An increase in intensity indicates membrane depolarization. **c** Mitochondrial ROS level as quantified from Mitosox™ red fluorescence. **d** Change in Ca²⁺-induced mPTP opening. **e** ATP content. **f** Cytochrome c oxidase activity. **g** Superoxide dismutase (SOD) activity. **h** Catalase activity. **i** Glutathione peroxidase (GPx) activity. **j** Reduced glutathione (GSH) content. **k** Schematic representation of the redox homeostasis mechanism by C-Mn₃O₄ NPs against H₂O₂ distress through mitochondrial protection. In bar plots data were expressed as Mean ± SD. Violins depict kernel density estimation of the underlying data distribution with the width of each violin scaled by the number of observations at that Y-value. Three lines (from the bottom to the top) in each violin plot show the location of the lower quartile (25th), the median, and the upper quartile (75th), respectively. The shaded area indicates the probability distribution of the variable. Individual data points are represented as colored circles ($N = 5$). One-way analysis of variance (ANOVA) followed by Tukey's post hoc multiple comparison test was performed for comparison among the groups. The numbers inside the plots indicate numerical p values. $p < 0.05$ is considered significant.

mitochondrial integrity. The mitochondrial membrane depolarization and subsequent opening of mPTP led to a significant fall in the cellular ATP content (Fig. 3e). In C-Mn₃O₄ NP pretreated cells, such loss in ATP content was not observed. The opening of mPTP, fall in $\Delta\Psi_m$ and ATP content cumulatively functions as a proapoptotic signal to initiate the cell death pathways, also reflected in the increased cytochrome-c oxidase activity (Fig. 3f). Superoxide dismutase (SOD), catalase (CAT), and glutathione peroxidase (GPx) constitute the intracellular antioxidant defense system that works in consort with

mitochondria[31–33]. The accumulation of highly reactive oxygen radicals causes damage to biomolecules in cells and alters enzyme activities[34–36]. Hence, we extended our study towards evaluating the effect of H₂O₂ and C-Mn₃O₄ NPs in the ROS regulatory network. H₂O₂ exposure significantly reduced the activity of SOD, CAT, and GPx resulting in a decrease of the reducing pool of cellular thiol constituents (e.g., GSH) (Fig. 3g–j). Pretreatment with C-Mn₃O₄ NPs significantly attenuated the damage. In cells treated with C-Mn₃O₄ NPs alone, none of the detrimental effects were observed.

Thus, our cellular studies indicate that C-Mn$_3$O$_4$ NPs possess the distinctive property of scavenging intracellular ROS, inhibiting apoptotic trigger, preventing loss of antioxidant enzymes, and maintaining high cell viability by acting as a protector of mitochondria, the master regulator of cellular redox equilibrium (Fig. 3k schematically summarizes the whole sequence).

**C-Mn$_3$O$_4$ NPs attenuate glomerular and tubulointerstitial damage in CKD mice.** There is always a gap in the efficacies of a pharmacological agent tested between cellular and animal model. Limited bioavailability, nonspecific biodistribution, or unwanted metabolism often restricts the in vivo use of a cytoprotective agent[37–39]. Thus, we evaluated the potential of C-Mn$_3$O$_4$ NPs in the treatment of cisplatin-induced C57BL/6j mice, a well-established animal model for testing therapeutic interventions against CKD[40–42]. As depicted in Fig. 4a, treatment with C-Mn$_3$O$_4$ NPs alone did not cause any mortality during the experimental period. While, chronic administration of cisplatin resulted in significant mortality (~40% compared to control), administration of C-Mn$_3$O$_4$ NPs in the cisplatin-intoxicated group significantly reduced the mortality (Hazards Ratio, HR (Log-rank): 2.62; 95% CI of HR: 1.166–4.75; log-rank $\chi2$ (Mantel–Cox): 5.23; df: 1; $P = 0.0222$) (Fig. 4a). The fourfold higher blood urea nitrogen (BUN) content (Fig. 4b), threefold higher GFR (Table 1), fourfold higher urinary albumin excretion (i.e., albuminuria) (Fig. 4c), and high urine albumin to creatinine ratio (ACR) (Table 1) along with significantly increased serum urea (Fig. 4d) and creatinine (Fig. 4e) illustrated induction of proteinuria and notable damage to the renal function of mice, the two hallmarks of CKD[43–45]. Treatment with C-Mn$_3$O$_4$ NPs (0.25 mg kg$^{-1}$ body weight (BW)) to the cisplatin intoxicated animals considerably reduced BUN, GFR, urinary albumin, ACR, serum urea, and creatinine (Fig. 4b–e and Table 1). Treatment with citrate (the functionalization group) was unable to reduce any of the aforementioned parameters (Supplementary Fig. S2), confirming the observed effects solely due to the conjugated nanomaterial. This may be due to the low dose of citrate (i.e., 0.25 mg kg$^{-1}$ body weight) used in the study. Such a low dose of citrate, a small molecule antioxidant having no catalytic activity, was not enough to prevent the oxidative damage caused by cisplatin. These results are well in agreement with the study by Kondo et. al.,[46] where the authors revealed that citrate alone has no nephroprotective action, but it enhances the protective effect of bismuth subnitrate against the *cis*-diamminedichloroplatinum induced nephrotoxicity. Cisplatin intoxication caused weight loss in mice (Fig. 4f), suggestive of the systemic toxicity that frequently arises in individuals receiving this anticancer drug. Animals treated with C-Mn$_3$O$_4$ NPs were capable of mitigating weight loss. Next, we examined the external morphology of isolated kidneys from each group. The kidneys from the cisplatin exposed group deviated from the usual darkish brown to a pale brown color with a rough and uneven surface (Fig. 4g). The kidney to body weight ratio (i.e, kidney index) was also significantly higher (i.e., edema) in cisplatin-treated animals (2.1 ± 0.2 compared to 1.5 ± 0.1 mg g$^{-1}$ of control, $p < 0.001$, one-way ANOVA, $F(3, 28) = 31.7$; Fig. 4h). Subsequent treatment with C-Mn$_3$O$_4$ NPs overturned the observed changes in morphology and kidney index.

Hematoxylin and eosin-stained kidney sections of the control and C-Mn$_3$O$_4$ NP treated groups showed normal histologic features (Fig. 4j) with negligible necrosis score (Fig. 4i). The kidney sections from cisplatin intoxicated mice displayed several pathological features of CKD like focal segmental as well as global glomerulosclerosis along with interstitial fibrosis, diffused thickening of the capillary walls, glomerular hyalinosis, dilated or

collapsed Bowman's space, and glomerular retraction (Fig. 4j). Tubular atrophy, dilation of cortical tubules, increased mesangial matrix, obliteration of capillaries, necrosis, vacuolization, and interstitial mononuclear infiltration were the other features observed in this group. Treatment with C-Mn$_3$O$_4$ NPs notably reduced focal glomerular necrosis (Fig. 4j). However, sparse tubular changes like vacuolization, dilation, mild mononuclear infiltration, and detachment of epithelial cells were observed in this group. Overall, C-Mn$_3$O$_4$ NPs were able to efficiently revert the marked detrimental changes in the renal architecture of CKD animals. The histological observations are quantitatively reflected in the necrosis score (Fig. 4i), glomerular injury score (GIS; Fig. 4k), and tubular injury score (TIS; Fig. 4l).

Previous studies and our histological observations suggested an association between renal fibrosis and CKD[47–49]. So, we measured the renal hydroxyproline content, a byproduct of collagen metabolism, and biochemical marker of fibrosis. The results indicate almost a threefold increase in the hydroxyproline content (1.43 ± 0.07 compared to 0.51 ± 0.03 mg gm$^{-1}$ tissue of control; $p < 0.001$, One-way ANOVA, $F(3, 16) = 378.7$) in the cisplatin intoxicated group (Table 1). In accordance with the histological findings, treatment with C-Mn$_3$O$_4$ NPs markedly reduced the hydroxyproline content (0.79 ± 0.06 mg gm$^{-1}$ tissue; $p < 0.001$ compared to cisplatin-treated ones, One-way ANOVA, $F(3, 16) = 378.7$), suggesting a decrease in fibrotic damage (Table 1).

**C-Mn$_3$O$_4$ NPs augment the intracellular antioxidant defense system.** Oxidative stress proved to be one of the major causes of cisplatin-induced nephrotoxicity[50–52]. Therefore, we tried to ascertain whether C-Mn$_3$O$_4$ NPs contributed to nephroprotection by ameliorating oxidative stress. Signs of ROS-mediated damage including lipid peroxidation (in terms of thiobarbituric acid reactive substances, TBARS), reduction in cellular GSH pool, and inhibition of antioxidant enzyme activities were estimated. Exposure to cisplatin markedly increased the level of TBARS (Fig. 5a) and oxidative glutathione along with a reduction in GSH concentration. Furthermore, it inhibited the antioxidant actions of enzymes like SOD, CAT, and GPx (Fig. 5b–d). The results, in consensus with our cellular studies, indicate that C-Mn$_3$O$_4$ NPs rescued the renal cells from detrimental pleiotropic effects of increased ROS while maintaining the normal signaling circuitry.

**C-Mn$_3$O$_4$ NPs reduce renal inflammation.** Macrophage infiltration in the kidney and subsequent rise in the plasma concentrations of pro-inflammatory cytokines like TNF-α are well-known features of CKD[53–55]. We found significant increases in plasma concentrations of TNF-α, IL-1β, and IL-6 in cisplatin-induced animals (Fig. 5e–g). Treatment with C-Mn$_3$O$_4$ NPs resulted in a notable decrease in cytokine levels. No difference was observed between the C-Mn$_3$O$_4$ NP treated and the control groups. Previous studies have demonstrated that direct adsorption of pro-inflammatory cytokines (e.g., TNF-α, ILs, etc.) by the nanoparticle surface can provide false positive or negative results about inflammatory responses[56,57]. Therefore, in order to further verify the findings of reduction in renal inflammation due to C-Mn$_3$O$_4$ NP treatment, we performed immunohistochemical (IHC) staining of the kidney sections with anti-CD68 antibodies, a well-known macrophage infiltration marker associated with the M1 macrophage phenotype. It is evident from Fig. 5h that the number of CD68 positive area is negligible in cisplatin+C-Mn$_3$O$_4$ NP co-treated animals compared to the cisplatin intoxicated animals, which showed pronounced staining of macrophage infiltrated area (marked in a dotted circle). The animals treated with C-Mn$_3$O$_4$ NPs alone showed characteristics similar to the

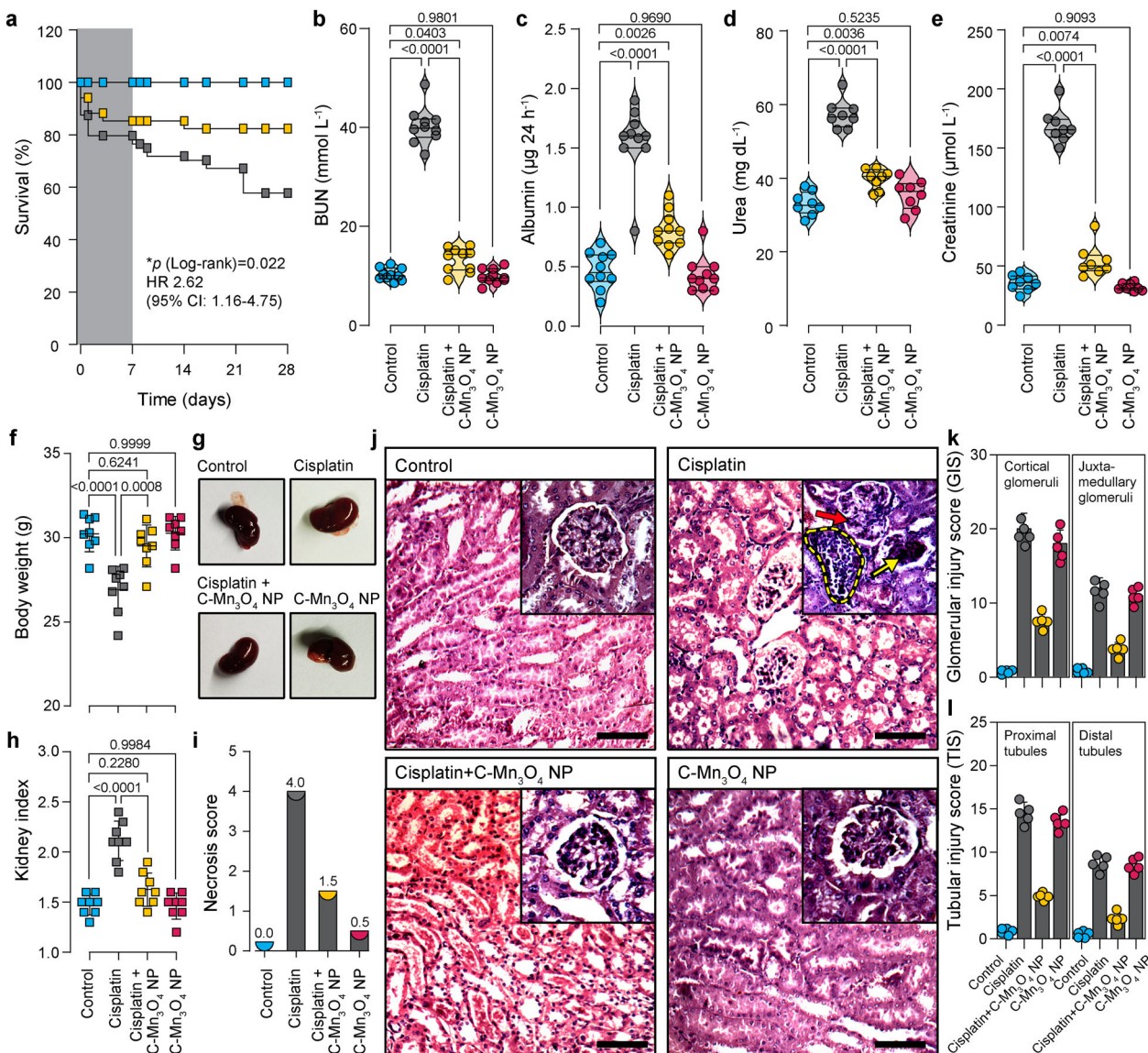

**Fig. 4 Efficacy of C-Mn$_3$O$_4$ NPs in the reversal of CKD in the animal model. a** Kaplan–Meier survival analysis curve. The darker shaded area represents the co-treatment period. **b** Blood urea nitrogen (BUN) content. **c** Urinary albumin excretion as an indicator of albuminuria, a hallmark of CKD. **d** Serum urea concentration. **e** Serum creatinine level. **f** Body weight at the end of the experimental period. **g** Photographs of kidneys incised after the experimental period. **h** Kidney index, defined as a kidney to body weight ratio (mg g$^{-1}$). **i** Necrosis score as per the observation of an expert clinical pathologist. **j** Hematoxylin and eosin-stained liver sections. Insets show a magnified image of a single glomerulus. Red arrow: segmental glomerulosclerosis; Yellow arrow: global glomerulosclerosis; Yellow dotted region: mononuclear infiltration. Scale bar: 20 μm. **k** Glomerular injury score (GIS). **l** Tubular injury score (TIS). In bar plots data were expressed as Mean ± SD. Violins depict kernel density estimation of the underlying data distribution with the width of each violin scaled by the number of observations at that Y-value. Three lines (from the bottom to the top) in each violin plot show the location of the lower quartile (25th), the median, and the upper quartile (75th), respectively. The shaded area indicates the probability distribution of the variable. Individual data points are represented as colored circles or squares (N = 10). One-way analysis of variance (ANOVA) followed by Tukey's post hoc multiple comparison test was performed for comparison among the groups. The numbers inside the plots indicate numerical p values. p < 0.05 is considered significant.

control ones (Fig. 5h). Therefore, the results together suggest that the observed reduction in the inflammatory markers by the administered nanoparticles happened due to modulation of the inflammatory cascade, not by direct adsorption of ILs on the surface of the particle.

**C-Mn$_3$O$_4$ NPs alleviate mitochondrial damage in CKD mice.** Considering the inevitable role of mitochondria in the pathogenesis of CKD[26,58–63] and the results of our in cellulo observations that C-Mn$_3$O$_4$ NPs protect mitochondria from H$_2$O$_2$ induced oxidative damage, we assessed the role of mitoprotection in the therapeutic

efficacy of C-Mn$_3$O$_4$ NPs in animals. Ca$^{2+}$-induced renal mPTP opening is one of the salient features of CKD[26,64]. Our data clearly show that the mitochondria isolated from the cisplatin intoxicated group were more sensitive towards Ca$^{2+}$ manifested into a sharp decrease in 540 nm absorbance (Fig. 6a). Treatment with C-Mn$_3$O$_4$ NPs inhibited mPTP opening and maintained membrane integrity. $\Delta\Psi_m$ and ATP content declined significantly as a result of cisplatin administration (Fig. 6b, c). These were accompanied by an increase in cytochrome c oxidase activity (Fig. 6d) and a reduction in dehydrogenase activity (Fig. 6e). The alterations were abrogated by C-Mn$_3$O$_4$ NP treatment. Mitochondrial parameters for the animals treated with C-Mn$_3$O$_4$ NPs alone showed no signs of toxicity and

**Table 1 Effect of C-Mn$_3$O$_4$ NPs on nephrotoxic biomarkers.**

| Group | GFR ($\mu$L min$^{-1}$ g$^{-1}$ BW) | Urine ACR | Creatinine clearance ($\mu$mol min$^{-1}$) | Uric acid (mg dL$^{-1}$) | Hydroxyproline (mg g$^{-1}$ tissue) |
|---|---|---|---|---|---|
| Control | 10.2 ± 1.5 | 0.34 ± 0.06 | 1.41 ± 0.08 | 1.2 ± 0.1 | 0.51 ± 0.03 |
| Cisplatin | 30.4 ± 4.1** | 5.62 ± 0.08** | 0.35 ± 0.04** | 2.6 ± 0.2** | 1.43 ± 0.07** |
| Cisplatin + C-Mn$_3$O$_4$ NPs | 14.1 ± 2.3* | 1.87 ± 0.09*,** | 0.92 ± 0.05*,** | 1.6 ± 0.1* | 0.79 ± 0.06*,** |
| C-Mn$_3$O$_4$ NPs | 9.8 ± 1.2* | 0.41 ± 0.05* | 1.38 ± 0.07* | 1.3 ± 0.1* | 0.42 ± 0.04* |

Data expressed as Mean ± SD ($N = 6$).
Hydroxyproline contents were measured from kidney homogenate.
One-way analysis of variance (ANOVA) followed by Tukey's post hoc multiple comparison test was performed for comparison among the groups.
GFR glomerular filtration rate, ACR albumin to creatine ratio.
*$p < 0.05$ compared to Cisplatin treated animals.
**$p < 0.05$ compared Control animals.

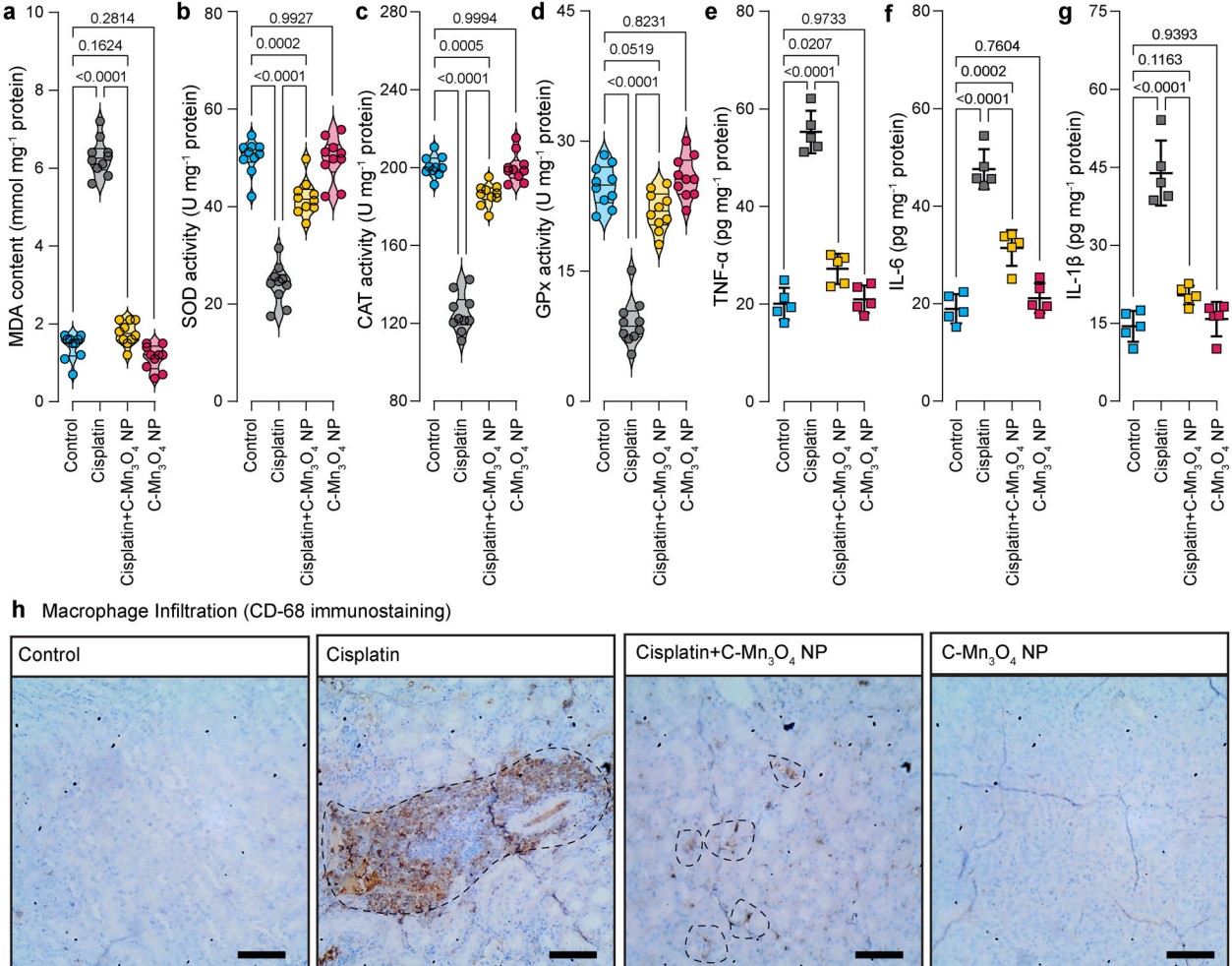

**Fig. 5 Effect of C-Mn$_3$O$_4$ NPs in the protection of intracellular redox regulatory network and inhibition of anti-inflammatory response in mice. a** Extent of lipid peroxidation (MDA, malonaldehyde content) was measured in terms of thiobarbituric acid reactive substances (TBARS). **b** Superoxide dismutase (SOD) activity. **c** Catalase activity. **d** Glutathione peroxidase (GPx) activity. **e** Tumor necrosis factor-α level. **f** Interleukin-1β level. **g** Interleukin-6 level. **h** Immunohistochemical analysis of kidney tissues for detection of inflammatory damages. Macrophages are stained with anti-CD-68 antibodies (brown). Scale bar: 20 μm. The dotted circles indicate the regions with high CD68 positivity (i.e., macrophage infiltration). MDA, SOD, CAT, and GPx were estimated from kidney homogenate. TNF-β, IL-1β, and IL-6 were measured from serum. Violins depict kernel density estimation of the underlying data distribution with the width of each violin scaled by the number of observations at that Y-value. Three lines (from the bottom to the top) in each violin plot show the location of the lower quartile (25th), the median, and the upper quartile (75th), respectively. The shaded area indicates the probability distribution of the variable. Individual data points are represented as colored circles ($N = 10$). One-way analysis of variance (ANOVA) followed by Tukey's post hoc multiple comparison test was performed for comparison among the groups. The numbers inside the plots indicate numerical $p$ values. $p < 0.05$ is considered significant.

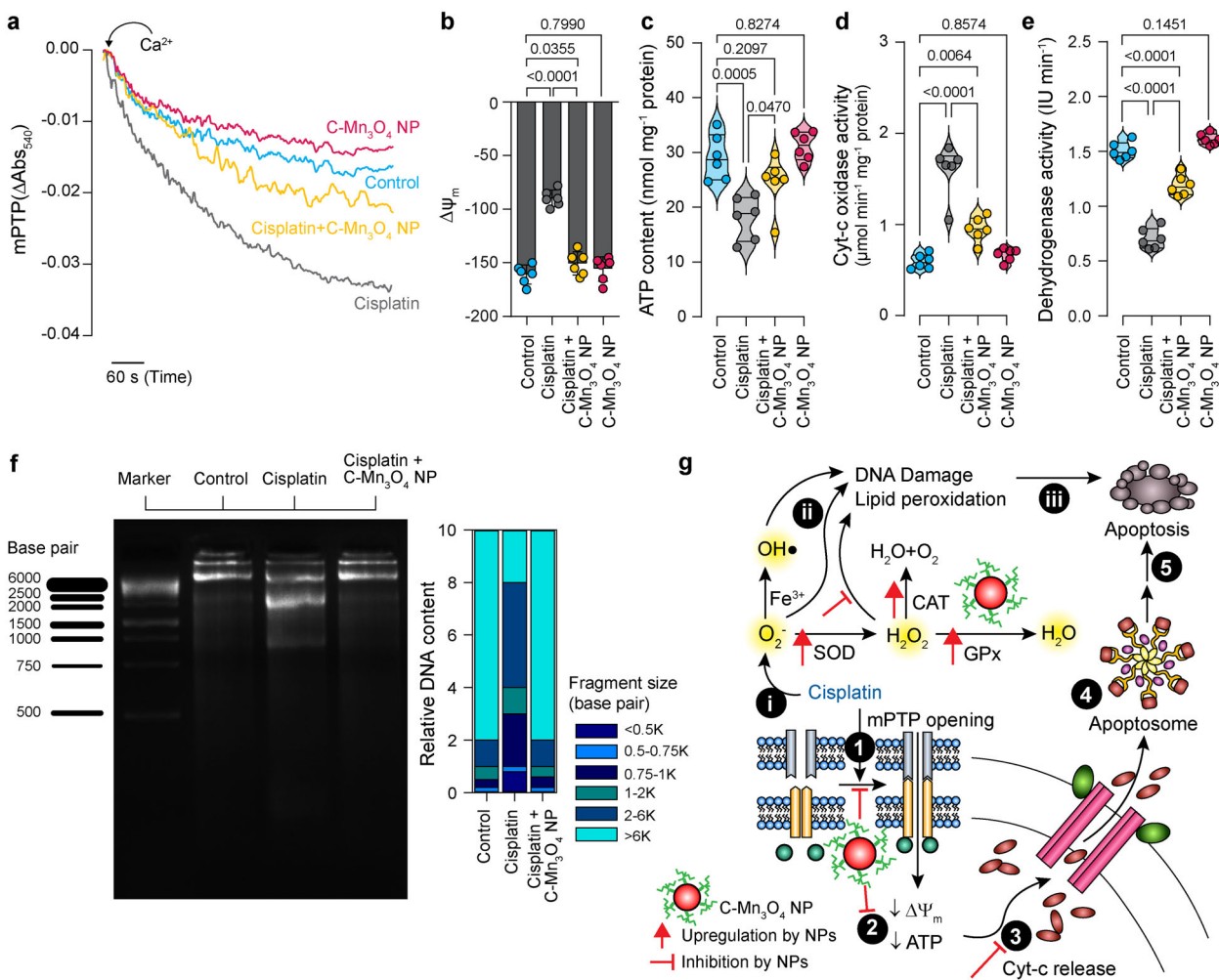

**Fig. 6 Efficacy of C-Mn₃O₄ NPs in the protection of mitochondria, the master redox regulator in mice. a** $Ca^{2+}$ induced mPTP opening measured by the decrease in 540 nm absorbance. **b** Mitochondrial membrane potential ($\Delta\psi_m$) estimated using JC-1 fluorescence. **c** ATP content. **d** Cytochrome c oxidase (complex IV in the electron transport chain, ETC) activity in isolated mitochondria. **e** Succinate dehydrogenase (SDH, complex II in ETC) activity in isolated mitochondria. **f** DNA fragmentation level as a result of oxidative damage measured using agarose gel electrophoresis. In cisplatin-induced CKD animals DNA ladder formation, indicative of apoptotic DNA fragmentation, is clearly visible. The corresponding stacked bar plot shows the relative abundance of different-sized fragmented DNA in relation to the total DNA content of the lane. **g** Schematic overview of the comprehensive molecular mechanism of action of C-Mn₃O₄ NPs as a redox medicine against cisplatin-induced CKD. The numbers in the black circles indicate the sequence of events. In bar plots data were expressed as Mean ± SD. Violins depict kernel density estimation of the underlying data distribution with the width of each violin scaled by the number of observations at that Y-value. Three lines (from the bottom to the top) in each violin plot show the location of the lower quartile (25th), the median, and the upper quartile (75th), respectively. The shaded area indicates the probability distribution of the variable. Individual data points are represented as colored circles ($N = 10$). One-way analysis of variance (ANOVA) followed by Tukey's post hoc multiple comparison test was performed for comparison among the groups. The numbers inside the plots indicate numerical $p$ values. $p < 0.05$ is considered significant.

were analogous to the control animals. Thus, cisplatin-induced renal damage triggered the opening of mPTP, the decline in $\Delta\psi_m$, and the induction of mitochondrial swelling that resulted in the release of cytochrome c in the cytosol leading to apoptosis. The ladder-like DNA fragmentation, a hallmark of apoptosis, was evident in the case of cisplatin-treated disease groups (Fig. 6f). Whereas, treatment with C-Mn₃O₄ NPs notably protected the mitochondria, inhibited cell death, and decreased the extent of DNA fragmentation. Figure 6g schematically illustrates the entire phenomena of redox-mediated nephroprotection by C-Mn₃O₄ NPs.

**Pharmacokinetics and biocompatibility of C-Mn₃O₄ NPs.** Over the years, pharmacokinetic (PK) studies have emerged as an integral part of drug development, especially for identifying the in vivo behavior of a drug. Therefore, we evaluated the time-

dependent plasma concentration profile of C-Mn₃O₄ NPs in order to figure out how the nanoparticles are absorbed into and eliminated from the bloodstream. Intraperitoneal (i.p.) administration of C-Mn₃O₄ NPs ($0.25\,mg\,kg^{-1}$ BW) to mice generated the plasma Mn concentration vs. time profile displayed in Fig. 7a. The PK parameters were calculated using a non-compartmental approach, yielding a maximum plasma concentration ($C_{MAX}$) of $3.1 \pm 0.2\,\mu g\,mL^{-1}$ at ($t_{MAX}$) $2.0 \pm 0.1$ h. The plasma area under the curve (AUC) was $20.5 \pm 1.8\,\mu g\,h\,mL^{-1}$ with a Clearance of $12.2 \pm 1.5\,L\,h^{-1}\,kg^{-1}$. The mean plasma concentration curve for C-Mn₃O₄ NPs (Fig. 7a) presented a two-peak (at ~2.0 and ~12.0 h) absorption phase. The first peak is due to the absorption of NPs from the peritoneal cavity to the blood. While, intestinal absorption, variable gastric emptying, enterohepatic recirculation, and distribution, and reabsorption are possible interpretations for the second peak (at ~12.0 h)[65]. Next, we evaluated the time-

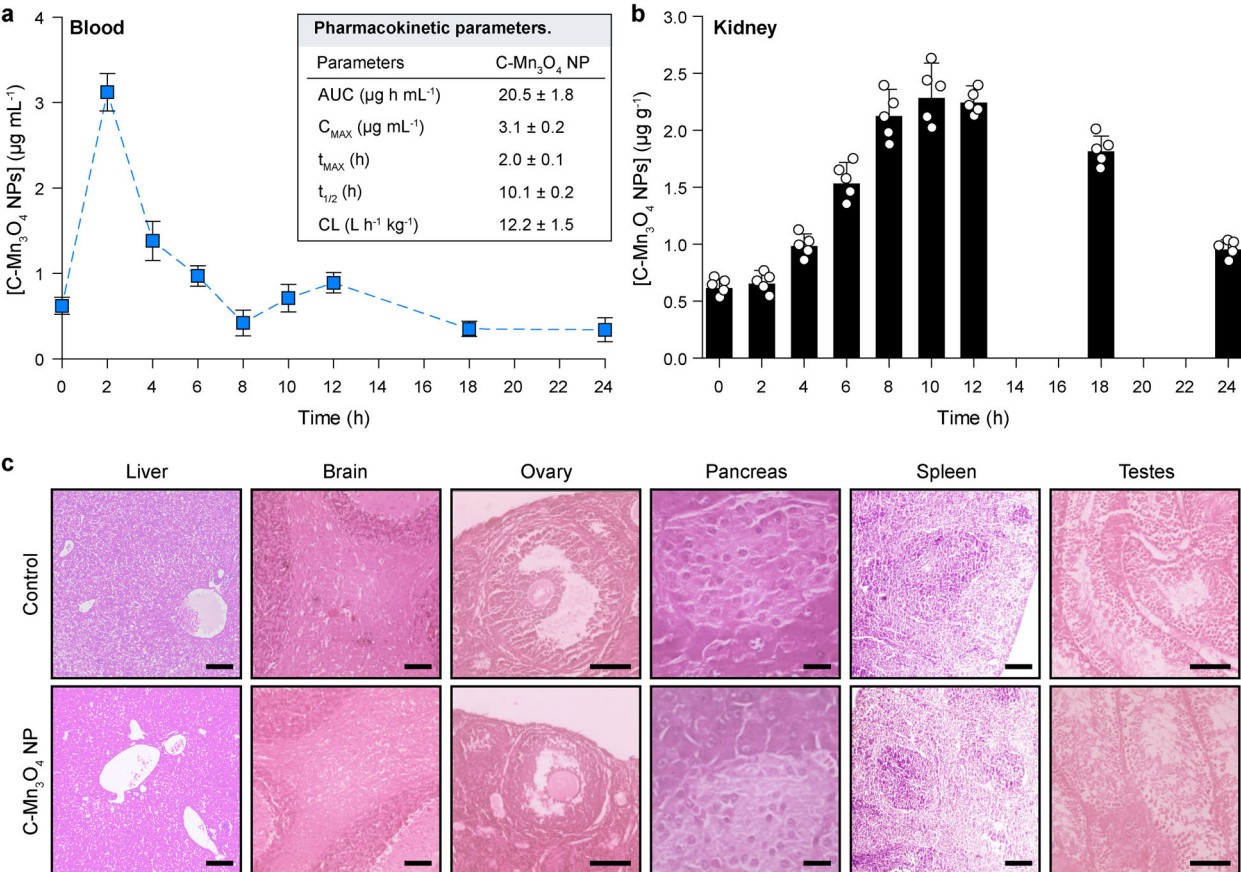

**Fig. 7 Pharmacokinetics (PK) and biocompatibility of C-Mn₃O₄ NPs. a** Plasma concentration-time profile following intraperitoneal administration of C-Mn₃O₄ NPs as measured using inductively coupled plasma-atomic emission spectroscopy (ICP-AES). Inset shows the PK parameters calculated using a non-compartmental approach. The dotted line is a guide to the eye. **b** Time-dependent accumulation and elimination of C-Mn₃O₄ NPs in the kidney. Data were normalized to wet kidney weight. **c** Micrographs of hematoxylin and eosin-stained sections of different organs after 1 month of treatment with the therapeutic dose (0.25 mg kg⁻¹ body weight, i.p.) of C-Mn₃O₄ NPs. Both control and treated animals maintained normal tissue architecture. Scale bars: liver-100 μm, brain-50 μm, ovary-20 μm, pancreas-50 μm, spleen-100 μm, testes-100 μm. Data were expressed as Mean ± SD. Individual data points are represented as white circles ($N = 5$).

dependent uptake and elimination profile of C-Mn₃O₄ NPs in the kidney up to 24 h (Fig. 7b). The manganese (i.e., C-Mn₃O₄ NP) content in the kidney increased exponentially reaching the maximum concentration of $2.28 \pm 0.31\ \mu g\ g^{-1}$ at ~10 h. The significantly increased concentration of nanoparticles in the kidney upon administration of C-Mn₃O₄ NPs clearly depicts its ability to enter the kidneys through the glomerular filtration barrier. Although the elimination phase started after 12 h, a sufficient quantity ($0.95 \pm 0.09\ \mu g\ g^{-1}$) of nanoparticles remained in the kidneys even after 24 h post administration.

Next, we investigated the effect of C-Mn₃O₄ NPs in other major organs in order to evaluate its biocompatibility. The hematoxylin and eosin-stained histopathological sections of the organs from C-Mn₃O₄ NP treated animals showed identical features to the organs collected from the control group (Fig. 7c). Livers of the nanoparticle-treated mice showed normal hepatic architecture with clearly detectable central vein, portal tracts, and hepatocytes arranged in cords with normal sinusoidal space. In the case of the pancreas, we found a normal pancreatic tissue structure comprising of pancreatic acini, pancreatic ducts, and β-cells islands. Nanoparticle treated spleen maintained ideal histological structure of white and red pulp. The testes of nanoparticle-treated animals did not show any remarkable structural changes. Their structure is comprised of well-organized seminiferous tubules and visible interstitial space. The maturation of spermatids were normal. The

histological analysis also revealed normal ovarian structure with follicles at different growth phases and observable luteum follicle. The brain structure was also typical for C-Mn₃O₄ NP treated mice with normal cerebellum consisting of the granular and molecular layer, and visible Purkinje cells.

## Discussion

In this study, we determined whether C-Mn₃O₄ NPs could be used as a redox medicine to treat CKD, an important clinical question considering the high prevalence of the disease and the nonavailability of effective medication. CKD is defined as the progressive and irreversible loss of renal function characterized by reduced glomerular filtration rate (GFR), increased urinary albumin excretion (albuminuria), or both[43,45,66]. Our results present evidence that treatment with C-Mn₃O₄ NPs significantly improved renal function, glomerular and tubulointerstitial injury, cellular antioxidant defense network in line with inhibition of pro-inflammatory immune response, and attenuation of mitochondrial dysfunction in response to the cisplatin toxicity. Our cellular and animal studies further enlightened the role of the unique mitoprotective as well as redox modulatory activity of C-Mn₃O₄ NPs in the therapeutic mechanism.

Several underlying factors played role in the selection of C-Mn₃O₄ NPs as the material of choice. These include the exciting redox modulatory properties (contrasting ability to

function as prooxidant as well as catalytic antioxidant, the fundamental feature of a redox medicine), biodegradability, aqueous solubility, low cost of the nanomaterial along with apparent non-toxicity (permissible limit ~12 mg day$^{-1}$), and abundance of manganese (Mn) as the catalytic metal centers or cofactors in several enzymes. Furthermore, previous reports[67] about the nephroprotective action of catalytic antioxidants encouraged us to evaluate the therapeutic potential of C-Mn$_3$O$_4$ NPs against CKD. This study provides prima facie indication that other biodegradable/organic nanomaterials with sufficient mitoprotective and redox modulatory activity could be used to treat CKD, provided they can enter and stay in the kidney for sufficient time to depict therapeutic effect. In this regard, it is worth mentioning here that the bioavailability of a nano-compound to the kidney could be enhanced using efficient carrier molecules. For example, the peptide amphiphile micelles (PAM) functionalized with the zwitterionic peptide ligand, (KKEEE)$_3$K, developed by Huang et. al.,[68] have the ability to cross the glomerular filtration barrier for the efficient delivery of therapeutic agents to kidney.

Mitochondria have long been recognized for their canonical roles in cellular respiration and energy production[26]. Recently, they have emerged as the master regulator of a spectrum of molecular pathways including biosynthesis of macromolecules, maintenance of cellular redox equilibrium, calcium homeostasis, inflammation, and cell death[69–73]. Thus, mitochondria are poised to play a pivotal role in the functioning of the kidney, an organ with high energy demand, and rich in mitochondria, second only to the heart[60]. Our findings that C-Mn$_3$O$_4$ NPs maintain cellular redox homeostasis through the prevention of mPTP opening and ATP depletion discloses a key redox-mediated nephroprotective mechanism. Virtually, the renal proximal tubules are exclusively dependent on ATP generated by mitochondrial oxidative phosphorylation and are therefore vulnerable to oxidative distress due to mitochondrial damage[50,74]. Cisplatin accumulates in mitochondria and reduces the activity of all four respiratory complexes (I–V) involved in the electron transport chain (ETC), thereby a surge in mitochondrial ROS formation takes place along with mPTP opening, membrane depolarization, and impairment in ATP production, leading to cell death[75,76]. The cytotoxic mechanism of cisplatin essentially mimics the pathogenesis of CKD, thus an efficient reversal of damage in this rodent model is supposed to reflect the possible effects of a compound in higher animals. Data from our cellular as well as animal studies provide sufficient evidence that C-Mn$_3$O$_4$ NPs ameliorate mitochondrial ROS surge, prevent loss of membrane potential, inhibit mPTP opening, and stops ATP depletion, thereby prevents mitochondrial dysfunction, cellular redox imbalance, and tubular or glomerular cell death. As a result, the markers of CKD i.e., increased BUN, plasma creatinine, serum urea, and GFR returns to homeostatic condition.

This study provides a piece of direct evidence that C-Mn$_3$O$_4$ NPs can scavenge ROS, particularly H$_2$O$_2$ the longest living one in the cellular milieu. It also proves the ability of the nanoparticles in the prevention of mPTP from opening and subsequent maintenance of mitochondrial structure and function. However, it is not clear whether ROS scavenging protects mitochondria or protection of mitochondrial integrity results in ROS depletion. Several studies have shown that a compound having antioxidant properties cannot be a sustainable therapeutic solution to oxidative-stress-related disorders, fundamentally due to its inability to regenerate after a single reaction[77–79]. In contrast, redox nanomaterials have the potential to reduce oxidative stress either by autocatalysis or by protecting cellular machineries that control redox homeostasis, until the nanomaterials dissolute in the physiological milieu. Considering the causal relationship between mitochondria and cellular redox homeostasis and the

efficacy of C-Mn$_3$O$_4$ NPs in the treatment of multifaceted diseases like CKD, we propose that both the mechanisms (i.e., ROS scavenging and mitochondrial protection) simultaneously take place.

The findings that C-Mn$_3$O$_4$ NPs can accelerate the revival of proximal tubule epithelium embodies a crucial nephroprotective function mediated by the nanoparticles. Kidneys show higher regenerative property following tubulointerstitial damage[50]. The proliferation of a subset of sublethally damaged, yet surviving, proximal tubule cells can contribute to the regenerative property of the kidney[80]. The acceleration of this process is sufficient to confer nephroprotection[81,82]. As revealed in our histological findings, the recovery rate of these cells (indicated by the structural integrity of the cellular architecture in hematoxylin and eosin-stained sections, and TIS scores) in C-Mn$_3$O$_4$ NP treated cisplatin exposed animals (Fig. 4j, k) was significantly higher and efficient than the auto-recovery (Fig. 4j, k). Several mechanisms can be proposed for the enhanced proliferation by C-Mn$_3$O$_4$ NPs. The restoration of structural and functional integrity of mitochondria and recovery of respiratory complexes may contribute towards the increased proliferation. It is well known that the mitochondrial ETC has a crucial role in cell proliferation through regulation of ATP generation, and supply of energy to proliferative pathways[83,84]. Earlier studies have demonstrated that mutations in ETC genes or the presence of ETC complex inhibitors cause a reduction in ATP synthesis, thus, obstructs progression through the cell cycle and proliferation[85–87]. Henceforth, it is reasonable to assume that the mitoprotective activity of C-Mn$_3$O$_4$ NPs have played a significant role in the revival of tubulointerstitial epithelial cells, in turn protecting the renal architecture. Additionally, ROS scavenging by C-Mn$_3$O$_4$ NPs may boost the proliferation because oxidative distress in proximal tubules causes cell cycle arrest and impedes cell-cycle progression[28].

The favorable PK properties and biocompatible nature further indicates the possibility of using C-Mn$_3$O$_4$ NPs as a nano-drug. Beyond size, the localization of a nano-drug in the target organ allows assessment of its therapeutic regimen. The direct evidence that the nanoparticles enter the kidney crossing the glomerular filtration barrier and reside there for a sufficient amount of time further justifies our claim that the recovery is due to the therapeutic action of the nanoparticles. It is worth mentioning here that in virtually all in vivo studies using nanomaterials, the bulk of the material ends up in the spleen and the liver. The same is true for C-Mn$_3$O$_4$ NPs. In one of our recent studies, we revealed that C-Mn$_3$O$_4$ NPs also have a tendency to accumulate in the liver if orally administered for a longer (90 days) period of time[88]. Although, identifying the particular cell types that internalizes C-Mn$_3$O$_4$ NPs to the kidney or liver, and understanding the exact mechanisms of internalization are intriguing, they fall beyond the scope of this study and remain an attractive area for further investigation.

The role of intracellular redox regulation through mitoprotection in the therapeutic action of C-Mn$_3$O$_4$ NPs opens up further avenues for the treatment of several unmet diseases like diabetic nephropathy, neurodegeneration (e.g., Parkinson's, Huntington's, Alzheimer's, multiple sclerosis), cardiovascular disorders, obesity, etc. where pathogenesis is very much dependent upon mitochondrial damage and associated redox imbalance[89–93]. Although in our study C-Mn$_3$O$_4$ NPs did not show any adverse effect, a detailed study on systemic toxicity, bio-distribution, PKs, and pharmacodynamics will greatly enhance the knowledge about its in vivo behavior. Furthermore, a detailed molecular study analyzing the genome and metabolome of C-Mn$_3$O$_4$ NP-treated animals may enlighten its ability to interfere in other pathogenesis pathways. As an outcome, the nano-drug could be repurposed for other therapies too.

## Conclusion

There are very few published articles in contemporary literature that utilize the promising redox regulatory approach for the treatment of chronic diseases like CKD. On the other hand, several CKDs are reported to be due to redox imbalance in mitochondria. Our study suggests that C-$Mn_3O_4$ NPs could be an efficient redox medicine to attenuate renal injury and tubuleintestinal fibrosis as evidenced by the improved renal functions, reduction in biochemical markers of nephrotoxicity, reduced fibrotic content, and downregulated proinflammatory cytokines. The molecular mechanism involves regulation of the redox balance through synchronization of the causal relationship between mitoprotection and ROS scavenging by C-$Mn_3O_4$ NPs. The findings highly suggest the translational potential of C-$Mn_3O_4$ NPs as a redox nanomedicine for treating CKD in the clinic.

## Methods

**Synthesis of C-$Mn_3O_4$ NPs**. A template or surfactant-free sol-gel method was followed for the synthesis of bulk $Mn_3O_4$ NPs at room temperature and pressure[21]. To functionalize the nanoparticles with ligand citrate, the as-prepared $Mn_3O_4$ NPs (~20 mg mL$^{-1}$) were mixed extensively with citrate (Sigma, USA) solution (pH 7.0, 0.5 M) for 15 h in a cyclomixer. The time for mixing was carefully adjusted to have a nanomaterial in the size range of 4–6 nm. Non-functionalized larger NPs were removed using a syringe filter (0.22 μm).

**Characterization techniques**. TEM and HRTEM images were acquired using an FEI TecnaiTF-20 field emission HRTEM (OR, USA) operating at 200 kV. Sample preparation was done by drop-casting of aqueous C-$Mn_3O_4$ NP solution on 300-mesh amorphous carbon-coated copper grids (Sigma, USA) and allowed to dry overnight at room temperature. XRD patterns were obtained by employing a scanning rate of 0.02 s$^{-1}$ in the $2\theta$ range from 10 to 80 by a PANalytical XPERT PRO diffractometer (Malvern, UK) equipped with Cu-Kα radiation operating at 40 mA and 40 kV. FTIR (JASCO FTIR-6300, Japan) was used to confirm the covalent attachment of the citrate molecules with the $Mn_3O_4$ NPs. For FTIR studies, powdered samples were blended with KBr powder and pelletized. KBr pellets were used as a reference to make the background correction.

**$H_2O_2$ scavenging by C-$Mn_3O_4$ NPs**. The ability of C-$Mn_3O_4$ NPs to prevent $H_2O_2$ mediated degradation of sodium-containing dye RB (Sigma, USA) was used as an indicator of its $H_2O_2$ scavenging activity. The addition of $H_2O_2$ (10 mM) in the aqueous solution of RB (3.5 μM) leads to decolorization of the dye reflected in a decrease in absorbance ($\lambda_{max}$ = 540 nm). The presence of C-$Mn_3O_4$ NPs (50 μg mL$^{-1}$) in the reaction mixture reduced degradation. All absorbance measurements were performed using Shimadzu UV-Vis 2600 spectrometer (Tokyo, Japan).

**Culture of human embryonic kidney cells (HEK 293)**. HEK 293 cells were maintained at 37 °C in 5% $CO_2$ in RPMI 1640 growth medium (Himedia, India) that contained 10% fetal bovine serum (Invitrogen, USA), L-glutamine (2 mM), penicillin (100 units mL$^{-1}$), and streptomycin (100 ng mL$^{-1}$) (Sigma, USA). Before experimentation, the cells were washed twice and incubated with RPMI 1640 medium (FBS, 0.5%) for 1 h and then treated as described in the figure legends.

We used $H_2O_2$ (Merck, USA) to induce oxidative damage to the cells considering the fact that $H_2O_2$ is one of the most predominant intracellular ROS involved in cellular signaling as well as oxidative damage. While, several other compounds can endogenously produce of free radicals (i.e., $CuSO_4$ or 3-amino-1,2,4-triazole), we did not use them in this study as none of them is naturally produced in the physiological system.

**Measurement of cell viability**. Cell viability was assessed by MTT and LDH assay. All cell lines were plated in 96-well plates at a density of $1 \times 10^3$ cells/well and cultured overnight at 37 °C. The treatments were performed as described in the figure legends. Next, MTT (5 mg mL$^{-1}$; Himedia, India) was added to each well, with a final concentration of ~0.5 mg mL$^{-1}$, and the cells were cultured for 4 h at 37 °C in a 5% $CO_2$ atmosphere. The resultant purple formazan was dissolved by the addition of 10% sodium dodecyl sulfate (Sigma, USA) and the absorbance was read at 570 and 630 nm using a microplate reader (BioTek, USA). LDH release was analyzed using a colorimetric LDH cytotoxicity assay kit (Himedia, India) following the manufacturer's instructions. Five independent experiments were performed in each case.

**Measurement of intracellular ROS**. The formation of intracellular ROS was measured with the DCFH2-DA method using both FACS and confocal microscopy. For FACS, cells were trypsinized, washed with 1X PBS, and stained with DCFH2-DA (15 μM; Sigma, USA) for 10 min at 30 °C in the dark after the treatments were completed. Ten thousand events were analyzed by flow cytometry (FACS Verse, Beckton Dickinson, SanJose, USA) and the respective mean fluorescence intensity (in FL1 channel, set with a 530/30 nm bandpass filter) values were correlated with the ROS levels. For confocal microscopy, 5000 cells were seeded and treated with agents described earlier. Post-treatment cells were stained with DCFH-DA (5 μM) at 37 °C in dark and images were acquired with a confocal microscope (Olympus IX84, Japan). All parameters (pinhole, contrast, gain, and offset) were held constant for all sections in the same experiment.

**Mitochondrial superoxide and $\Delta\Psi_m$ detection**. After treatment with the drug or vehicle, mitochondrial superoxide production was visualized using MitoSOX$^{TM}$ Red (Thermofisher, USA), a mitochondrial superoxide indicator. The $\Delta\Psi_m$ in intact cells was assessed by confocal microscopy using Rh 123 (Sigma, USA). Cells were loaded with MitoSOX$^{TM}$ Red (0.5 μM) or Rh123 (1.5 μM) for 10 min at 37 °C and imaged using a confocal microscope (Olympus IX84, Japan). All parameters (pinhole, contrast, gain, and offset) were held constant for all sections in the same experiment. ImageJ (http://imagej.nih.gov/ij/) was used to quantify area normalized fluorescence intensities from the confocal images.

**Animals and treatment**. Healthy nondiabetic C57/6j mice of both sexes (8–10 weeks old, weighing 27 ± 2.3 g) were used in this work. Animals were maintained in standard, clean polypropylene cages (temperature 21 ± 1 °C; relative humidity 40–55%; 1:1 light and dark cycle). Water and standard laboratory pellet diet for mice (Saha Enterprise, Kolkata, India) were available *ad libitum* throughout the experimental period. All mice were allowed to acclimatize for 2 weeks before the treatment. The guideline of the Committee for the Purpose of Control and Supervision of Experiments on Animals (CPCSEA), New Delhi, India, was followed and the study was approved by the Institutional Animal Ethics Committee (Ethical Clearance No. - 05/S/UC-IAEC/01/2019).

The mice were randomly divided into five groups ($n = 16$ for each group): (1) control; (2) cisplatin; (3) cisplatin + C-$Mn_3O_4$ NPs; (4) C-$Mn_3O_4$ NPs; and (5) cisplatin + citrate. The experimental model of CKD was established according to the previous description. In brief, for induction of CKD, we used 8 mg kg$^{-1}$ BW cisplatin (i.p.) in each alternative day for 28 days. After induction, we treated C-$Mn_3O_4$ NPs at 0.25 mg kg$^{-1}$ BW (i.p.) for another 28 days. There was an overlap of 7 days between induction and treatment. Citrate (Sigma, USA) was used at a dose of 0.25 mg kg$^{-1}$ BW for 28 days. All doses were finalized based on reported literature and pilot experimentation. As citrate treatment did not improve kidney function, data were not represented in the main manuscript.

**Biochemical evaluations**. Whole blood samples from treated mice were collected from retro-orbital sinus plexus and centrifuged at $2000 \times g$ for 20 min to separate the serum. Urine samples were collected in metabolic cages during 24-h fasting conditions. Biochemical evaluations were performed using commercially available kits (Autospan Liquid Gold, Span Diagnostic Ltd., India) following the protocol described by respective manufacturers. GFR was estimated by the determination of urinary excretion of fluorescein-labeled inulin (FITC–inulin, Sigma, USA).

**Histological examination**. After incision, tissues from randomly selected mice were fixed with 4% paraformaldehyde, embedded in paraffin, and cut into a 5 μm thick section. After de-waxing and gradual hydration with ethanol (Merck, USA), the sections were stained with hematoxylin and eosin (SRL, India). The sections were then observed under an optical microscope (Olympus, Tokyo, Japan). It is noteworthy to mention here that the histopathologist was blinded to the treatment groups while scoring and evaluating the samples. For immunohistochemistry, kidney sections were incubated for 60 min with rat anti-mouse CD68 antibody (Santa Cruz Biotechnology, India) followed by a 30 min incubation with 10 mg mL$^{-1}$ HRP-conjugated rabbit anti-rat secondary antibody (Santa Cruz Biotechnology, India). After detection of peroxidase activity with 3-amino-9-ethylcarbazole (Sigma, USA), sections were counterstained with Mayer's hematoxylin (SRL, India).

**Renal hydroxyproline measurement**. For the measurement of renal hydroxyproline content, a previously described method was used[32]. In brief, snap-frozen kidney specimens (200 mg) were weighed, hydrolyzed in HCl (6 M; Merck, USA) for 12 h at 100 °C. Next, they were oxidized with Chloramine-T (SRL, India). Next, Ehrlich reagent (Sigma, USA) was added which resulted in the formation of a chromophore. Absorbance was measured at 550 nm. Data were normalized to kidney wet weight.

**Renal homogenate preparation**. Samples of kidney tissue were collected, homogenized in cold phosphate buffer (0.1 M; pH 7.4), and centrifuged at 10,000 rpm at 4 °C for 15 min. The supernatants were collected for further experimentation.

**Assessment of lipid peroxidation and hepatic antioxidant status**. The supernatants obtained in the previous stage were used to measure the activity of SOD, CAT, GPx, and GSH as well as the content of lipid peroxidation (MDA). Lipid peroxidation was determined in TBARS formation using a reported procedure[31].

SOD (Sigma, MO, USA), CAT (Abcam, Germany), and GPx activities (Sigma, MO, USA) were estimated using commercially available test kits following protocols recommended by respective manufacturers. The renal GSH level was determined by the method of Ellman with trivial modifications[94].

**Mitochondria isolation and mitochondrial function determination**. Mitochondria were isolated from mouse kidneys following the method described by Graham[95] with slight modifications. In brief, kidneys were excised and homogenized in a kidney homogenization medium containing 225 mM D-mannitol, 75 mM sucrose, 0.05 mM EDTA, 10 mM KCl, and 10 mM HEPES (pH 7.4). The homogenates were centrifuged at $600 \times g$ for 15 min and the resulting supernatants were centrifuged at $8500 \times g$ for 10 min. The pellets were washed thrice and resuspended in the same buffer. All procedures were performed at 4 °C.

Mitochondrial function was evaluated by determining $\Delta\Psi_m$ using JC-1 (Sigma, MO, USA), ATP production (Abcam, Germany), and the activities of mitochondrial complexes succinate dehydrogenase and cytochrome c oxidase. mPTP opening was measured in terms of mitochondrial swelling by monitoring the decrease in absorbance at 540 nm after the addition of $CaCl_2$ (100 mM).

**Agarose gel electrophoresis for DNA fragmentation**. Total renal DNA was isolated following a standard procedure[96]. DNA fragmentation was assessed using agarose gel electrophoresis. In a typical procedure, renal DNA (5.0 µg) was loaded on 1.5% agarose gel stained with ethidium bromide. Electrophoresis was carried out for 2 h at 90 V, and the resultant gel was photographed under UV transillumination (InGenius 3 gel documentation system, Syngene, MD, USA).

**Pharmacokinetics, cellular and kidney uptake**. For PK studies, animals were administered with C-$Mn_3O_4$ NPs (0.25 mg kg$^{-1}$ BW; i.p.). Then, blood and kidney were collected at different time points, and Mn contents were estimated using ICP-AES (ARCOS-Simultaneous ICP Spectrometer, SPECTRO Analytical Instruments GmBH, Germany). The open acid digestion method was employed for the preparation of samples. In brief, dried tissue samples (in liquid nitrogen) were dissolved in a 3:2:1 mixture of $HNO_3$ (Merck, Germany), $H_2SO_4$ (Merck, Germany), and $H_2O_2$ (Merck, Germany), and heated at 150 °C until only a residue remained. Then, the residues were diluted to 10 mL using Mili-Q water. Data were normalized to kidney wet weight.

For cellular internalization studies, HEK 293 cells were treated with different concentrations of C-$Mn_3O_4$ NPs and then subjected to ICP-AES following a similar procedure as described above. Data were normalized to the wet weight of the cells.

**Statistics and reproducibility**. Kaplan-Meier survival curves were used to illustrate mortality. Differences in survival between groups were assessed by the log-rank test with multiple pair-wise comparisons performed using the Mantel–Cox method. All quantitative data are expressed as Mean ± Standard Deviation (SD) unless otherwise stated. One-way analysis of variance (ANOVA) followed by Tukey's post hoc multiple comparison test was performed for comparison between multiple groups. Beforehand, the normality of each parameter was checked by normal quantile–quantile plots. Sample size in our animal studies were determined following the standard sample sizes previously been used in similar experiments as per relevant literature. Designated sample size (in figure legends) always refers to biological replicates (independent animals). GraphPad Prism v8.0 (GraphPad Software) and Sigmaplot v14.0 (Systat Software, Inc.) were used for statistical analysis. Origin Pro 8.5 (OriginLab Corporation, MA, USA) was used for fitting the data. For all comparisons, a $p$ value <0.05 was considered statistically significant.

**Reporting summary**. Further information on research design is available in the Nature Research Reporting Summary linked to this article.

## Data availability

All essential data are provided in the manuscript. The datasets generated and analyzed during this study to support the findings are available in a DOI-minting online open access repository, figshare, with the identifier (https://doi.org/10.6084/m9.figshare.14995122)[97]. Any remaining information can be obtained from the corresponding author upon reasonable request.

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

## Acknowledgements

M.D. thanks University Grants Commission (UGC), Govt. of India for Junior Research Fellowship. S.K.P. thanks the Indian National Academy of Engineering (INAE) for the Abdul Kalam Technology Innovation National Fellowship, INAE/121/AKF. The authors thank the DBT (WB)-BOOST scheme for the financial grant, 339/WBBDC/1P-2/2013. The authors would like to thank Sophisticated Analytical Instrument Facility SAIF), Indian Institute of Technology (IIT) – Bombay, India for carrying out the ICP-AES studies and Oncquest Laboratories Ltd., India for helping in IHC studies.

## Author contributions

A.A., M.B. and S.K.P. designed the experiments. A.A., S.M. and P.B. did synthesis, characterization, and in vitro studies. T.C. and M.B. performed cellular studies. A.A., S.D., S.M. and M.D. conducted animal experiments. A.K.D. performed the histological studies and analysis. R.G. performed the biodistribution, and cellular uptake studies during revision. A.A., H.A., J.T.A., A.S., S.A.A., M.B. and S.K.P. discussed and analyzed the results. A.A. wrote the manuscript and all authors contributed towards the writing of the final version.

## Competing interests

The authors declare no competing interests.
