## [Peer Review File · Communications Biology]

Reviewers' comments:

Reviewer #1 (Remarks to the Author):

This is a well-written article regarding the development of nanoparticles that can decrease oxidative species in chronic kidney disease. However, much of the nanocharacterization and in vivo assessment is limited including:

- 1) biodistribution/PK studies: are the nanoparticles, even though small and should go to the kidneys through the glomerular filtration barrier, the authors do not show this.
- 2) if targeting the kidneys, are they staying in the kidneys, and if so, for how long? this will allow assessment of therapeutic regimen; more broadly, beyond size, how is the nanoparticles staying in the kidneys?
- 3) why did the citrate not have a therapeutic effect? shouldn't it be cleared renally? or is it getting cleared too fast to cause a therapeutic response? this needs to be validated.
- 4) articles regarding kidney targeting by small nanoparticles such as by Huang, Yi et al bioTM 2020 need to be included.
- 5) to assess compatibility of nanoparticles and lack of adverse effects, histology for all other organs need to be shown
- 6) macrophages and lack of inflammation needs to be verified and quantified by IHC using CD68, flow cytometry, etc
- 7) why this type of nanoparticle is useful for CKD is not addressed; is it biodegradable? why not other degradable materials/organic nanomaterials?

Reviewer #2 (Remarks to the Author):

Review "Redox Nanomedicine Cures Chronic Kidney Disease by Mitochondrial Reconditioning"
Adhikari et al.

In this paper, Adhikari et al. use the free radical scavenging abilities of Mn₃O₄ nanoparticles to ameliorate oxidative damage in both an in vitro (HEK cell culture) and in vivo, murine disease model. This work is important in translating the benefit of nanoparticle therapies across biological test beds. As the authors discuss, few studies demonstrate efficacy both in vitro and in vivo. While the data provided by the authors is compelling and fits nicely with their interpretations of the data, I feel the work could be strengthened with some key experiments measuring MnOx content in cells and tissues.

Major Concerns

The major concern I have is that the authors provide no measurement of tissue or cellular content of MnOx nanoparticles in the study. The biological consequences of exposure to the nanomaterial can only be understood in terms of content. While the particles assuredly entered the cell given the mitochondrial measurements, it is critical to know the actual content (ppm/g wet weight). Parsing apart beneficial versus toxic effects across studies requires knowledge of tissue/cellular content. Without this information, we cannot determine if the biological effects are simply dose dependent or related to subtle difference in nanoparticle composition. It should not be difficult to incubate the HEK in the concentrations used in the study and determine content with ICP-MS. The same could be done for the in vivo studies if tissue is still available. If enough data points were available, then one could plot dose, content and biological impact (ROS levels, MTT, etc). If you did this for the in vitro and in vivo work you could relate the in vitro dosing to the in vivo, which would be very useful. In addition, there should be some comment that in virtually all in vivo studies using nanomaterials the bulk of the material ends up in the spleen and the liver.

Minor Concerns

Line Comment

1 'Ameliorate' is a better word choice than 'Cures'

54 'indigenously' I am not sure why you are using this word

56 'animal model of oxidative injury'

85 'Apathy' seems like an awkward word choice.

112-114 Redox state can drive valence/activity of metal oxides but so can pH. pH in biological systems varies extra/intracellular and within organelles.

126 Maybe comment on role of protein corona as well.

164 1mM H₂O₂ seems much higher than needed

174 are their compounds that can induce endogenous production of free radicals as opposed to using H₂O₂

219 Figure 2 shows marked oxidant effects of nanoparticle alone on ROS, LDH . While it is important that MnOx mitigated changes with H₂O₂ challenge, did you perform specific comparisons between control and nanoparticles alone as part of the post-hoc analysis?

228 'unwarranted' awkward word use

226 it may be useful to begin the section by describing the effects of the MnOx nanoparticle alone on your measures. Then move on to the treatment groups. Also, in this section do you have any idea which hepatic cell types accumulated the material (Ito cells, Kupfer cells, hepatocytes etc).

238 'treatment with MnO₃O₄' plus CIS

254 were individuals scoring blinded to the treatment?

263 how were animals selected for this analysis? Was it based on some biological measurement (i.e. highest GPx, MDA etc. from each treatment group).

295 any idea if direct adsorption of ILs by particles could reduce serum levels as opposed to reduction of inflammatory cascades

307 consider 'abrogated' as opposed to upended.

315 this figure has no data on MnOx nanoparticles alone and I am not sure why.

339 consider accumulates as opposed to 'accrues'

351 you cannot say this provides direct evidence unless you actually measure the presence of the material tissues/cells.

356-358 I am not sure this statement is true for redox nanomaterials; they are effective at reducing redox stress until they dissolve/degrade. This type of regenerative, auto-catalysis is what makes them such good antioxidants.

361-366 this section could be shortened.

372-374 I am not sure which data supports this statement? Describe/tell the readers which metrics are relevant to the statement.

Reply to the Reviewers

We are thankful to the learned reviewers for the careful evaluation of our manuscript. Their comments and suggestions were very helpful in enhancing the quality of the work. In the following section, we have addressed all the concerns of the learned reviewers point by point.

Reviewer 1:

Comment 1. This is a well-written article regarding the development of nanoparticles that can decrease oxidative species in chronic kidney disease. However, much of the nano characterization and in vivo assessment is limited including:

Response 1. The authors would like to thank the learned reviewer for the kind appreciation and valuable comments to improve the quality of the manuscript. In the following section, we have tried to address all the concerns raised by the learned reviewer point by point.

Query 1. Biodistribution/PK studies: are the nanoparticles, even though small and should go to the kidneys through the glomerular filtration barrier, the authors do not show this.

Reply 1. The authors would like to appreciate the concern of the learned reviewer. As per the kind suggestion of the learned reviewer, we have performed detailed PK study along with distribution of the nanoparticles in the kidney. The significantly increased concentration of nanoparticles in the kidney upon administration of C-Mn₃O₄ NPs clearly depicts its ability to enter kidney through the glomerular filtration barrier. We have addressed the issue in the revised manuscript. (Page 12, Line 362, Figure 7a & 7b)

Query 2. If targeting the kidneys, are they staying in the kidneys, and if so, for how long? This will allow assessment of therapeutic regimen; more broadly, beyond size, how is the nanoparticles staying in the kidneys?

Reply 2. The authors would like to appreciate the concern of the learned reviewer. As per the kind suggestion of the learned reviewer, we have assessed time dependent (up to 24 hours) distribution of the nanoparticles (by measuring Mn content) in the kidney using inductively couple plasma atomic emission spectroscopy (ICP-AES). The results show that the nanoparticle concentration in the kidney increased in a time dependent manner, and between 8-18 hours after single intra-peritoneal injection the accumulation was significantly high (C_{max} at 10 hours) (Figure 7b, revised manuscript). The elimination phase started after 12 hours. And even after 24 hours, a significant amount of nanoparticles remained in the kidney. We have addressed the issue in the revised manuscript. (Page 12, Line 376, Figure 7b)

Query 3. Why did the citrate not have a therapeutic effect? Shouldn't it be cleared renally? or is it getting cleared too fast to cause a therapeutic response? This needs to be validated.

Reply 3. The authors would like to thank the learned reviewer for pointing out the issue. As correctly pointed out, in our study citrate showed no therapeutic effect against CKD. This may be due to the low dose administered (i.e., 0.25 mg kg⁻¹ body weight). As citrate is a small molecule antioxidant having no catalytic activity, such low amount was not enough to prevent the oxidative damage caused by cisplatin. The results are well in agreement with the study by Kondo *et. al.*, (Ref: *Cancer Chemother. Pharmacol.* (2004) 53: 33–38) where the authors showed that citrate has no nephro-protective action, but it enhances the protective effect of bismuth subnitrate against the cis-diamminedichloroplatinum induced nephrotoxicity. Furthermore, the use of sodium citrate in the treatment of nephrotoxicity is sparse in the contemporary literature. It is generally used for the treatment of metabolic acidosis. However, as discussed above, the dose used is several multitudes higher (i.e., 80 mg kg⁻¹ body weight for humans, which is roughly 1600 mg kg⁻¹ body weight for mice) than that used in our study. We have addressed the issue in the revised manuscript. (Page 9, Line 271)

Query 4. Articles regarding kidney targeting by small nanoparticles such as by Huang, Yi et al bioTM 2020 need to be included.

Reply 4. The authors would like to thank the learned reviewer for the kind suggestion. We have discussed the topic of kidney targeting by small nanoparticles in the revised manuscript and included the recommended citations. (Page 14, Line 425)

Query 5. To assess compatibility of nanoparticles and lack of adverse effects, histology for all other organs need to be shown.

Reply 5. The authors would like to thank the learned reviewer for pointing out the issue. As per the kind advice of the learned reviewer we have evaluated the sub-chronic (30 days) toxicity of the nanoparticle at the administered dose and included the histopathological observations for other major organs (Figure 7c, revised manuscript). It has to be noted that, no observable toxic effects were found in nanoparticle treated group and the results show the biocompatibility of it. We have addressed the issue in the revised manuscript. (Page 13, Line 384)

Query 6. Macrophages and lack of inflammation needs to be verified and quantified by IHC using CD68, flow cytometry, etc.

Reply 6. The authors are thankful to the learned reviewer for the kind suggestion. As per the kind advice of the learned reviewer, we have verified the lack of inflammation using IHC

(CD-68 immunostaining) and included the results in the revised manuscript. (Page 11, Line 330, Figure 5h)

Query 7. Why this type of nanoparticle is useful for CKD is not addressed; is it biodegradable? Why not other degradable materials/organic nanomaterials?

Reply 7. The authors would like to thank the learned reviewer for pointing out the issue. As correctly pointed out by the learned reviewer, other degradable materials/organic nanomaterials having the mitoprotective and redox modulatory activities could be used to treat CKD, provided they can enter and stay in kidney for sufficient time to depict therapeutic effect. We selected nano-sized Mn_3O_4 as our compound of interest, encouraged by the apparent non-toxicity (permissible limit $\sim 12 \text{ mg day}^{-1}$), abundance of manganese (Mn) as the catalytic metal center or cofactor in several enzymes, and the occurrence of spontaneous comproportionation and disproportionation of surface ions (between +2, +3 and +4 states of Mn) depending upon microenvironment. Moreover, the nanoparticle is low-cost, biodegradable and have significant catalytic antioxidant properties. We have discussed the topic in the revised manuscript. (Page 14, Line 413)

Reviewer 2:

Comment 1. In this paper, Adhikari et al. use the free radical scavenging abilities of Mn₃O₄ nanoparticles to ameliorate oxidative damage in both an in vitro (HEK cell culture) and in vivo, murine disease model. This work is important in translating the benefit of nanoparticle therapies across biological test beds. As the authors discuss, few studies demonstrate efficacy both in vitro and in vivo. While the data provided by the authors is compelling and fits nicely with their interpretations of the data, I feel the work could be strengthened with some key experiments measuring MnOx content in cells and tissues.

Response 1. The authors would like to thank the learned reviewer for appropriately summarizing our work and kindly appreciating it. The authors would also like to thank the learned reviewers for the valuable comments to improve the quality of the manuscript. In the following section, we have tried to address all the concerns raised by the learned reviewer point by point.

Major Concerns

Query 1. The major concern I have is that the authors provide no measurement of tissue or cellular content of MnOx nanoparticles in the study. The biological consequences of exposure to the nanomaterial can only be understood in terms of content. While the particles assuredly entered the cell given the mitochondrial measurements, it is critical to know the actual content (ppm/g wet weight). Parsing apart beneficial versus toxic effects across studies requires knowledge of tissue/cellular content. Without this information, we cannot determine if the biological effects are simply dose dependent or related to subtle difference in nanoparticle composition. It should not be difficult to incubate the HEK in the concentrations used in the study and determine content with ICP-MS. The same could be done for the in vivo studies if tissue is still available. If enough data points were available, then one could plot dose, content and biological impact (ROS levels, MTT, etc.). If you did this for the in vitro and in vivo work you could relate the in vitro dosing to the in vivo, which would be very useful.

Reply 1. The authors would like to thank the learned reviewer for the kind suggestion. As correctly pointed out, the cellular/tissue content of nanoparticles is one of the important deciding factors in their biological effects. As per the kind suggestion of the learned reviewer, we measured concentration (i.e., dose) dependent cellular uptake of the nanoparticles using inductively couple plasma atomic emission spectroscopy (ICP-AES) (Figure 2f, revised manuscript). We further correlated the biological impact (MTT and ROS levels) with administered dose, and intracellular nanoparticle content (Figure 2g, revised manuscript). We have also assessed the PK (Figure 7a, revised manuscript) and time dependent (up to 24 hours) distribution of the nanoparticle in the kidney. The results show that the nanoparticle concentration in the kidney increased in a time dependent manner, and between 8-18 hours after single intra-peritoneal injection the accumulation was significantly high (C_{\max} at 10 hours) (Figure 7b, revised manuscript). The elimination phase started after 12 hours. We have addressed the issue in the revised manuscript. (Page 7, Line 206; Page 12, Line 376)

Query 2. In addition, there should be some comment that in virtually all in vivo studies using nanomaterials the bulk of the material ends up in the spleen and the liver.

Reply 2. The authors would like to thank the learned reviewer for pointing out the issue. As per the kind suggestion of the learned reviewer, we have discussed the topic in the revised manuscript. (Page 16, Line 93)

Minor Concerns

Query 3. 1 'Ameliorate' is a better word choice than 'Cures'.

Reply 3. The authors would like to thank the learned reviewer for the kind suggestion. We have changed the word in the revised manuscript. (Page 1, Line 1)

Query 4. 54 'indigenously' I am not sure why you are using this word.

Reply 4. The authors would like to thank the learned reviewer for pointing out the issue. As per the kind suggestion of the learned reviewer, we have removed the word in the revised manuscript. (Page 2, Line 54)

Query 5. 56 'animal model of oxidative injury'.

Reply 5. The authors would like to thank the learned reviewer for the kind suggestion. We have modified the statement in the revised manuscript. (Page 2, Line 56)

Query 6. 85 'Apathy' seems like an awkward word choice.

Reply 6. The authors would like to thank the learned reviewer for pointing out the issue. As per the kind suggestion of the learned reviewer, we have modified the word in the revised manuscript. (Page 3, Line 86)

Query 7. 112-114 Redox state can drive valence/activity of metal oxides but so can pH. pH in biological systems varies extra/intracellular and within organelles.

Reply 7. The authors would like to thank the learned reviewer for pointing out the issue. As correctly pointed out, the pH in the biological system can modulate the valence/activity of the metal oxides. We have discussed the topic in the revised manuscript. (Page 4, Line 113)

Query 8. 126 Maybe comment on role of protein corona as well.

Reply 8. The authors are thankful to the learned reviewer for the kind suggestion. We have discussed the role of protein corona in the revised manuscript. (Page 5, Line 129)

Query 9. 164 1mM H₂O₂ seems much higher than needed.

Reply 9. The authors would like to thank the learned reviewer for pointing out the issue. The concentration of H₂O₂ stock solution was 1 mM, however in the cell culture medium the final concentration was 100 μM. We have modified the unintentional mistake in the revised manuscript. (Page 6, Line 167)

Query 10. 174 are their compounds that can induce endogenous production of free radicals as opposed to using H₂O₂.

Reply 10. The authors appreciate the concern of the learned reviewer. There are several other compounds that can enhance the endogenous production of free radicals as opposed to using H₂O₂. For example, 500 μM CuSO₄ or 20 mM 3-amino-1,2,4-triazole. However, none of them is produced in physiological system. On the other hand, H₂O₂ is one of the most dominant intracellular ROS involved in cellular signaling as well as oxidative damage. Therefore, we used it for our cellular studies. We have addressed the issue in the revised manuscript. (Page 20, Line 561)

Query 11. 219 Figure 2 shows marked oxidant effects of nanoparticle alone on ROS, LDH. While it is important that MnOx mitigated changes with H₂O₂ challenge, did you perform specific comparisons between control and nanoparticles alone as part of the post-hoc analysis?

Reply 11. The authors would like to thank the learned reviewer for pointing out the issue. We have included the post-hoc analysis (one-way ANOVA followed by Tukey's multiple comparison test) between control and nanoparticles alone and included the results in the revised manuscript. (Page 6, Line 183; Page 7, Line 198)

Query 12. 228 'unwarranted' awkward word use.

Reply 12. The authors would like to thank the learned reviewer for pointing out the issue. As per the kind suggestion of the learned reviewer, we have modified the word in the revised manuscript. (Page 8, Line 252)

Query 13. 226 it may be useful to begin the section by describing the effects of the MnOx nanoparticle alone on your measures. Then move on to the treatment groups. Also, in this section do you have any idea which hepatic cell types accumulated the material (Ito cells, Kupfer cells, hepatocytes etc).

Reply 13. The authors would like to thank the learned reviewer for the kind suggestion. We have started and proceed with the section as advised by the learned reviewer. We are thankful to the reviewer for pointing out the issue of hepatic uptake. However, the sole aim of this study was to evaluate the potential of C-Mn₃O₄ NPs in the treatment of CKD. Therefore we have not studied the uptake by hepatic cells. We will definitely do a detailed study regarding the topic as suggested by the learned reviewer. We have addressed the issue in the revised manuscript. (Page 9, Line 256)

Query 14. 238 'treatment with MnO₃O₄' plus CIS.

Reply 14. The authors would like to thank the learned reviewer for pointing out the issue. As per the kind suggestion of the learned reviewer, we have corrected the group in the revised manuscript. (Page 9, Line 266)

Query 15. 254 were individuals scoring blinded to the treatment?

Reply 15. The authors would like to appreciate the kind concern of the learned reviewer. The histopathologist was blinded to the treatment groups while scoring and evaluating the samples. We have included the information in the 'Methods' section of the revised manuscript. (Page 22, Line 627)

Query 16. 263 how were animals selected for this analysis? Was it based on some biological measurement (i.e. highest GPx, MDA etc. from each treatment group).

Reply 16. The authors would like to appreciate the kind concern of the learned reviewer. It has to be noted that the animals were randomly selected for this analysis. We have included the information in figure legend and methods section. (Page 22, Line 624)

Query 17. 295 any idea if direct adsorption of ILs by particles could reduce serum levels as opposed to reduction of inflammatory cascades.

Reply 17. The authors would like to thank the learned reviewer for pointing out the issue. We have performed immunohistochemistry study for directly assessing macrophage infiltration by staining the kidney sections with anti-CD68 antibodies. It has to be noted that, our immunohistochemistry studies also suggest significant reduction in the macrophage inflammation which is only possible through modulation of inflammatory cascade, not by

direct adsorption of ILs by the particle. We have addressed the issue in the revised manuscript. (Page 11, Line 327, Figure 5h)

Query 18. 307 consider 'abrogated' as opposed to upended.

Reply 18. The authors would like to thank the learned reviewer for the kind suggestion. We have changed the word in the revised manuscript. (Page 12, Line 352)

Query 19. 315 this figure has no data on MnOx nanoparticles alone and I am not sure why.

Reply 19. The authors would like to thank the learned reviewer for pointing out the issue. The data were unintentionally missed during compilation of the figures and due to wrong axis selection. We have included the data in the revised manuscript. (Figure 7)

Query 20. 339 consider accumulates as opposed to 'accrues'.

Reply 20. The authors would like to thank the learned reviewer for the kind suggestion. We have changed the word in the revised manuscript. (Page 15, Line 440)

Query 21. 351 you cannot say this provides direct evidence unless you actually measure the presence of the material tissues/cells.

Reply 21. The authors would like to thank the learned reviewer for pointing out the issue. As per the kind suggestion of the learned reviewer, we have measured the accumulation of nanoparticle inside the tissue and included the information in the revised manuscript. (Page 12, Line 376)

Query 22. 356-358 I am not sure this statement is true for redox nanomaterials; they are effective at reducing redox stress until they dissolve/degrade. This type of regenerative, auto-catalysis is what makes them such good antioxidants.

Reply 22. The authors would like to thank the learned reviewer for pointing out the issue. As per the kind suggestion of the learned reviewer, we have modified the statement for better clarity. (Page 15, Line 459)

Query 23. 361-366 this section could be shortened.

Reply 23. The authors would like to thank the learned reviewer for the kind suggestion. We have shortened the indicated section in the revised manuscript. (Page 15, Line 462)

Query 24. 372-374 I am not sure which data supports this statement? Describe/tell the readers which metrics are relevant to the statement.

Reply 24. The authors would like to thank the learned reviewer for pointing out the issue. We have modified the statement and indicated the metrics for better clarity. (Page 16, Line 471)